# Large Language Models as Generalizable Policies for Embodied Tasks

**Andrew Szot, Max Schwarzer, Harsh Agrawal, Bogdan Mazoure, Walter Talbott**
**Katherine Metcalf, Natalie Mackraz, Devon Hjelm, Alexander Toshev**
Apple

## Abstract

We show that large language models (LLMs) can be adapted to be generalizable policies for embodied visual tasks. Our approach, called Large LAnguage model Reinforcement Learning Policy (LLaRP), adapts a pre-trained frozen LLM to take as input text instructions and visual egocentric observations and output actions directly in the environment. Using reinforcement learning, we train LLaRP to see and act solely through environmental interactions. We show that LLaRP is robust to complex paraphrasings of task instructions and can generalize to new tasks that require novel optimal behavior. In particular, on $1,000$ unseen tasks it achieves $42\%$ success rate, $1.7$x the success rate of other common learned baselines or zero-shot applications of LLMs. Finally, to aid the community in studying language conditioned, massively multi-task, embodied AI problems we release a novel benchmark, Language Rearrangement, consisting of $150,000$ training and $1,000$ testing tasks for language-conditioned rearrangement. Video examples of LLaRP in Language Rearrangement and the code are at https://llm-rl.github.io.

## 1 Introduction

Large Language Models (LLMs), characterized as billion-parameter models trained on enormous amounts of text data, have demonstrated unprecedented language understanding capabilities. Furthermore, LLMs have demonstrated powerful capabilities beyond core language understanding problems, such as dialog systems (Thoppilan et al., 2022; Glaese et al., 2022), visual understanding problems (Alayrac et al., 2022; Li et al., 2023b; Peng et al., 2023; Koh et al., 2023), reasoning (Wei et al., 2022; Lewkowycz et al., 2022), code generation (Chen et al., 2021b), embodied reasoning (Driess et al., 2023), and robot control (Ahn et al., 2022). These capabilities often emerge in a zero-shot fashion, without dedicated training data for each capability, indicating that LLMs contain knowledge general and broad enough to apply to numerous domains. Furthermore, these capabilities emerge despite that the input and output spaces in these domains are oftentimes not naturally expressed in language, e. g. images as inputs, and robot commands as outputs.

A key objective in Embodied AI is generalizable decision-making that can transfer to novel tasks, so it is natural to ask if the generalization abilities of LLMs can be incorporated into embodied problems. Existing advances in using LLMs for Embodied AI have relied on static expert datasets (Driess et al., 2023; Brohan et al., 2023), which requires prohibitively large and expensive amounts of diverse expert data. Conversely, Embodied AI simulators enable agents to learn from an environment through direct interaction, exploration, and reward feedback (Kolve et al., 2019; Szot et al., 2021; Li et al., 2023a). However, the generalization capabilities of such agents to a large number of new embodied tasks are not on par with the aforementioned domains.

LLMs have been shown to be applicable in online settings when the control domain is that of natural language, e.g., Reinforcement Learning from Human Feedback (RLHF) for multi-turn dialog applications (Ouyang et al., 2022). In this work, we successfully show that LLMs can be adapted via reinforcement learning as a vision-language policy for problems in embodied AI, using a method we call Large LAnguage model Reinforcement learning Policy (LLaRP). We demonstrate advanced capabilities on a diverse set of rearrangement tasks, where the input and output domains aren't just language (see Fig. 1) In particular, we demonstrate the following three contributions:

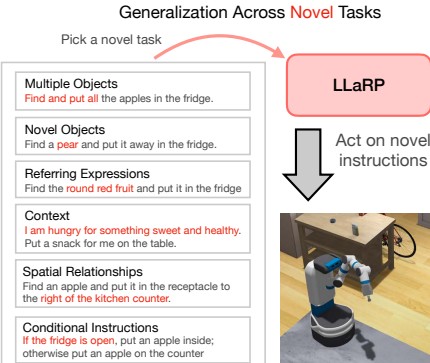

Figure 1: We demonstrate that by utilizing Reinforcement Learning together with a pre-trained LLM and maximizing only sparse rewards, we can learn a policy that generalizes to novel language rearrangement tasks. The method robustly generalizes over unseen objects and scenes, novel ways of referring to objects, either by description or explanation of an activity; and even novel descriptions of tasks, including variable number of rearrangements, spatial descriptions, and conditional statements.

First, we show that using a pre-trained and frozen LLM as a Vision-Language Model (VLM) policy with learned input and output adapter layers results in a policy that exhibits strong generalization capabilities. We train this policy using online RL and measure generalization along two axes:

- *Paraphrastic Robustness* (PR): the agent produces the same optimal behavior under linguistic variations of an instruction where the "intention" of the instruction does not change. This includes novel ways of describing the same behavior or novel ways of referring to a seen object.
- *Behavior Generalization* (BG): the agent solves tasks that require novel optimal behavior. This means the desired behavior outcome is distinct from those seen during training. For example, act on new types of or combinations of objects, new types of combined behaviors (e.g., finding "all" of something) or new logical conditions (e.g., "if" statements).

LLaRP is thoroughly evaluated on over $1,000$ unseen tasks spanning the above axes and attains $42\%$ success rate, compared to $25\%$ for an LSTM-based policy and $22\%$ for zero-shot applications of LLMs. Our approach outperforms all baselines both when it is instructed in novel ways as well as when tasked to perform unseen behaviors. Further, we demonstrate that the LLaRP LLM-based policy gives a non-trivial boost on another domain, Atari, compared to a Transformer baseline.

We demonstrate that, when the agent has access to the world knowledge encoded in an LLM, RL exhibits various forms of *training efficiencies*. For one, LLM-based models exhibit better *sample efficiency* than other common architectures in both basic PPO RL and continual learning settings (training the model on downstream tasks beyond the training tasks). Further, we show that LLaRP is more *efficient with what supervision is needed* than commonly used imitation learning.

Finally, in order to facilitate the above contributions and promote future work studying generalization in Embodied AI, we introduce the Language Rearrangement task which includes 150,000 distinct language instructions, each with automatically generated rewards. The large number of diverse tasks brings the system closer to real-world setups where agents should be able to do anything and everything, and pushes the limits on performance. For generalization evaluation, we define splits that stress test the system on PR- and BG-types of generalization.

## 2 RELATED WORK

Prior work has demonstrated large language models (LLMs) can be used as zero-shot policies for interactive decision-making tasks, without task-specific training, in settings where the states and action spaces are both text-based (Zeng et al., 2022; Shah et al., 2023; Huang et al., 2022; Liang et al., 2023; Huang et al., 2023b; Wu et al., 2023a; Silver et al., 2023; Wang et al., 2023). Namely, Code as Policies (Liang et al., 2023) relies on perception modules not available in our setting. Furthermore, LLMs can be adapted to text-based decision making by fine-tuning with a standard language modeling objective (Chalvatzaki et al., 2023). In addition, it has been shown that LLMs can be used to provide rewards or useful high-level goals for learning policies (Du et al., 2023; Hu & Sadigh, 2023; Wu et al., 2023b) in text-based or human-interaction settings.

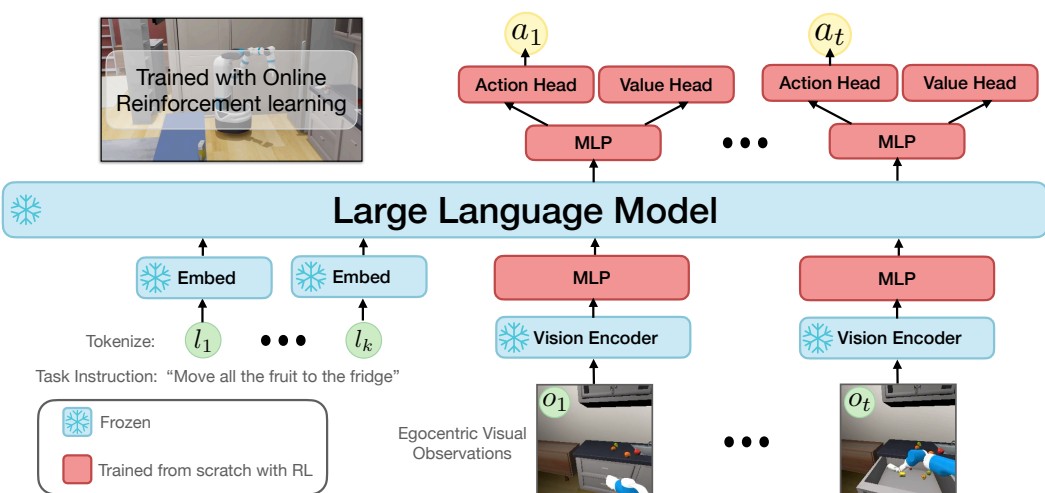

Figure 2: LLaRP architecture. The bottom of the figure shows the policy inputs including the task instruction and the egocentric visual RGB frames from the current time step to the start of the episode. These are encoded using the LLM embeddings or a vision encoder. The embeddings are input to a pre-trained LLM. The hidden outputs are then projected to action and value predictions. The entire system learns from online RL, where the action output module and observation encoder MLP are the only trained components and all other components are frozen.

It has been shown that agents can learn tasks specified by natural language. Shridhar et al. (2022); Team et al. (2021); Abramson et al. (2022); Yu & Mooney (2022); Lynch et al. (2023); Liu et al. (2022); Shridhar et al. (2023); Brohan et al. (2022); Ha et al. (2023) learn language conditioned policies through imitation learning on a dataset of expert trajectory and language pairs. Namely, Perceiver-actor Shridhar et al. (2023) requires offline data for imitation learning while LLaRP learns from interaction through RL. Xiao et al. (2022); Yu et al. (2023); Myers et al. (2023) augment the paired dataset to increase the quantity and diversity of data for imitation learning. Open-vocabulary mobile manipulation (OVMM, Yenamandra et al., 2023) demonstrated open-vocab rearrangement capabilities in pick-and-place settings, while other works demonstrate open-vocab object manipulation capabilities in navigation settings (Gadre et al., 2023; Shah et al., 2023; Huang et al., 2023a; Bolte et al., 2023).

Works in Vision Language Models (VLMs) have combined pre-trained LLMs with visual reasoning (Alayrac et al., 2022). Works have imbued VLMs with 3D spatial information (Hong et al., 2023; Jatavallabhula et al., 2023; Ha & Song, 2022), usually in a static setting without environment interaction. Like our work, PaLM-E (Driess et al., 2023) and RT-2 (Brohan et al., 2023) extend VLMs to interactive decision-making in visual embodied environments, but using high-quality expert data.

To our knowledge, no prior work demonstrates that LLMs can be used as vision-language policies in online RL problems to improve generalization. Like our work, Carta et al. (2023) also adapts an LLM with online RL, but it does so in environments with textual observations and text actions, while we adapt LLMs for vision-language policies with online RL. Furthermore, we demonstrate these capabilities over a diverse set of evaluations, both in terms of linguistic (paraphrastic generalization) and target behavior (behavior generalization) variations. Our Language Rearrangement extends beyond the scope of prior work using LLMs in embodied settings by requiring complex manipulation, navigation, and exploration. The tasks contain 18x more instructions involving more complex rearrangement concepts and object interactions than the benchmark ALFRED (Shridhar et al., 2020), one of the more common language-conditioned Embodied AI benchmarks.

## 3 METHOD

Our method, *Large LAnguage model Reinforcement learning Policy* (LLaRP), adapts pre-trained LLMs to operate in embodied multi-modal decision-making settings. We show how to modify existing LLMs for embodied settings and train a policy for embodied tasks, leading to an agent with significantly improved generalization capabilities to new language instructions.

Our problem can be formulated as a Partially-Observable Markov Decision Process (POMDP), defined as a tuple $(\mathcal{S}, \mathcal{O}, \mathcal{A}, \mathcal{P}, \mathcal{R}, \rho_0, \gamma)$ for underlying state space $\mathcal{S}$, observation space $\mathcal{O}$, action space $\mathcal{A}$, transition distribution $\mathcal{P}$, reward function $\mathcal{R}$, initial state distribution $\rho_0$ and discount factor $\gamma$. In our setting, $\mathcal{O}$ are egocentric high-dimensional visual observations, such as a robot's RGB camera which only observes part of the scene. We consider the extension of including a goal distribution $\mathcal{G}$ and the case where the reward is formulated as $\mathcal{R}(s, g)$ for $s \in \mathcal{S}$ and $g \in \mathcal{G}$. We seek to learn a goal-conditioned policy $\pi(a|o, g)$ mapping from observation $o$ and goal $g$ to an action $a$ that maximizes the sum of discounted rewards $\mathbb{E}_{s_0 \sim \rho_0, g \sim \mathcal{G}} \sum_t \gamma^t \mathcal{R}(s_t, g)$. Furthermore, we seek to learn a policy that generalizes and achieves high rewards for goal distributions $\mathcal{G}'$ not seen during training. Specifically, we consider goals specified as natural language instructions and want policies to generalize to new distributions of natural language instructions.

We utilize large language models (LLMs), which are large auto-regressive text prediction models. Given text represented as a sequence of tokens $l$, the LLM is trained to predict each token in that sequence conditioned on all prior tokens $\pi^{\text{LLM}}(l_{K+1} \mid l_1, \dots, l_K)$. Since an embodied agent policy needs to consume visual observations $\mathcal{O}$ and predict actions $\mathcal{A}$, both of which are not language tokens, we strip away the input and output layers of the LLM. In particular, the LLM input layer encodes the text tokens producing vector embeddings $e_k = E^T(l_k) \in \mathbb{R}^D$, while the output layer classifies words.

After we strip away the input and output layers, we call the resulting network an *LLM backbone* and denote it by $\psi^{\text{LLM}}(e_1, \dots, e_K) \in \mathbb{R}^D$. This backbone consumes a sequence of $D$-dim. token embeddings and produces a $D$-dim. token embedding.

### 3.1 LARGE LANGUAGE MODEL REINFORCEMENT LEARNING POLICY ARCHITECTURE

LLaRP, visualized in Figure 2, has two input types. First, it is conditioned on a goal $g = (l_1, \dots, l_k) \in \mathcal{G}$ expressed as language. This goal can be embedded using the language encoder $E_\theta^T$ into a sequence of $D$-dim. vectors. Second, it consumes visual observations $o_1, \dots, o_t$ during the policy rollout which are encoded using a separate learnable *observation encoder module* $E_\phi^V : \mathcal{O} \mapsto \mathbb{R}^D$. The observation encoder module consists of a vision encoder that produces a visual embedding and an MLP network that projects the visual embedding to the language model token embedding dimension. The encoded text and visual observations create a $k + t$ length sequence of $D$-dimension embeddings, which are input to the LLM backbone $\psi_\theta^{\text{LLM}}$, as defined above.

To decode an action as output, we employ a learnable *action output module* $D_\omega : \mathbb{R}^D \mapsto D(\mathcal{A})$ that converts the output of the LLM backbone to a distribution over actions from $\mathcal{A}$. With the two additional adapter modules $D_\omega$ and $E_\phi^V$ we are able to adapt the LLM to take as input goal specification and visual observations up to time $t$ in order to output action at time $t$:

$$\pi^{\text{LLaRP}}(a_t|o_1, \dots, o_t, g) = D_\omega(\psi_\theta^{\text{LLM}}(E_\theta^T(l_1), \dots, E_\theta^T(l_k), E_\phi^V(o_1), \dots, E_\phi^V(o_t)).$$

The action output module is an MLP that predicts a distribution over environment actions. We exclude the first $k$ outputs from $\psi^{\text{LLM}}$ that correspond to the language task specification tokens. The hidden outputs corresponding to observations at each time step are fed through the action modeling MLP to produce the distribution over actions. The action output module also predicts the value estimate used for the reinforcement learning update. More details about the visual encoder and action output module are in Appendix C.1.

### 3.2 LARGE LANGUAGE MODEL REINFORCEMENT LEARNING POLICY TRAINING

We train LLaRP using only reinforcement learning (Sutton & Barto, 2018). Specifically, we train with DD-PPO (Wijmans et al., 2019), an adaptation of PPO (Schulman et al., 2017) for multi-GPU training. LLaRP collects experience interactively by rolling out its policy auto-regressively to generate actions to take in the environment. For the PPO update, we sample minibatches from the collected data and compute action log probabilities in parallel for the PPO update.

During training, we freeze the LLM backbone and the visual encoder. A frozen visual encoder helps maintain visual features that can generalize to different environments (Majumdar et al., 2023). A frozen LLM backbone helps maintain language reasoning capabilities, which can be lost during fine tuning (Alayrac et al., 2022). For more details about the method, refer to Appendix C.1.

| | Dataset Name | Instruction Example | Description |
|---|---|---|---|
| **Paraphrasic Robustness** | Train | *Find an apple and put it away in the fridge.* | Instructions used to train the agent. |
| | New Scenes | *Find an apple and put it away in the fridge* | Same instructions, but in new scenes. All other datasets are in new scenes. |
| | Instruction Rephrasing | *In the fridge, stow an apple.* | Swap the order that nouns appear in the instruction and substitute synonyms for verbs. |
| | Referring Expressions | *Find the round red fruit and put it in the fridge* | Refer to objects by their visual appearance. |
| | Spatial Relationships | *Find an apple and put it in the receptacle to the right of the kitchen counter.* | Refer to receptacles indirectly by their location relative to other receptacles. |
| | Context | *I am hungry for something sweet and healthy. Put a snack for me on the table.* | Describe a situation where a particular object fits the context. |
| | Irrelevant Instruction Text | *There is a pear on the counter. Find an apple and put it away in the fridge.* | Instructions that include irrelevant context. |
| **Behavior Gen.** | Multiple Rearrangements | *Find an apple, pear, and banana and put them away in the fridge.* | Rearrange 3 objects (2 max during training). |
| | Novel Objects | *Find a orange and put it away in the fridge* | New entity / instruction pairs. |
| | Multiple Objects | *Put all the apples in the fridge.* | Find all of an object. |
| | Conditional Instructions | *If the fridge is open, find an apple and put it there. Otherwise put an apple on the table.* | Adjust behavior based on if the conditional statement is true. |

Table 1: Evaluation datasets. The datasets with unseen instructions are divided into two categories: paraphrastic robustness which tests if the agent can produce the same behavior under linguistic variation and behavior generalization where the agent has to demonstrate a new type of behavior. The red text highlights the concept in the instruction distinct from the training distribution that emphasizes the dataset evaluation purpose. See Appendix B.5 for more details.

## 4 LANGUAGE REARRANGEMENT PROBLEM

To study generalization properties across a large number of language conditioned tasks we introduce a novel problem called "Language Rearrangement". Language Rearrangement strives to cover a large number of tasks of home environment tasks, such as, "*Bring a mug to the couch.*", "*Store all the fruit in the fridge.*", or "*I am hungry, bring something from the kitchen to the table.*". This problem space extends the Rearrangement task (Batra et al., 2020) by defining 150,000 training and 1,000 testing tasks, and providing a textual instruction for each one. The tasks require an agent to generalize to a variety of unseen instruction concepts requiring multiple object interactions (picking, placing, opening, closing), searching for objects and logical reasoning (e.g. "if" statements).

### 4.1 TASK DEFINITION

In Language Rearrangement, an agent starts in an unseen house and is tasked to execute a common household activity, that reduces to moving objects from specified start positions to desired goal positions. The agent is provided with a natural language instruction specifying the desired goal state. We generate instructions at scale using instruction templates, instruction template rephrasings, scene-grounded entity enumeration, and a solver feasibility validation check. A sparse reward is provided for successfully completing the entire task or a subtask. Completing the instructions requires the agent to explore. For example, to "*Put all the dishes in the sink if the sink is empty*" requires the agent to explore to find all the dishes in the house. While exploring, the agent needs to detect when it sees a plate and then pick it up. The instructions vary in what information is revealed to the agent (such as the starting positions of objects), how many objects to rearrange, and logical concepts such as "for all", "exists", conditionals, swapping, and removals. See complete task details in Appendix B. We compare Language Rearrangement to other benchmarks in detail in Appendix B.7.

The agent has to perform the tasks entirely from onboard sensing capabilities and without any pre-built maps, object 3D models, or other privileged information. A simulated Fetch robot (Fetch Robotics, 2020) senses the world through a $256 \times 256$-pixel head-mounted RGB camera, robot joint positions, an indicator of whether the robot is holding an object or not, and base egomotion giving the relative position of the robot from the start of the episode. The agent interacts with the world

via a mobile base, 7DoF arm, and suction grip attached to the arm. Language Rearrangement is implemented in Habitat 2.0 (Szot et al., 2021). When the policy selects a valid low-level skill policy that can be executed in the current state, the simulation is kinematically updated with the skill post conditions.

We supplement all methods with low-level skills and focus on high-level decision-making. Language Rearrangement poses the challenge of long-horizon tasks spanning tens of thousands of low-level control. Even single-object mobile pick and place tasks are challenging for end-to-end methods (Berges et al., 2023; Huang et al., 2023c). Yet hierarchical methods where a high-level policy selects from fixed skills is effective for rearrangement tasks (Gu et al., 2022). Therefore like other rearrangement works (Szot et al., 2023), we train a high-level policy to select low-level policies. There are 70 skills for the high-level policy to choose between at every step. These skills include picking objects by name, placing on receptacles by name, navigating to receptacles, and opening and closing receptacles by name. Example skills include pick(apple), place(sink), navigate(left counter), and open(fridge). See Appendix B.3 for details.

## 4.2 GENERALIZATION TEST DATASETS

Language Rearrangement seeks to evaluate how well agents can complete language instructions during evaluation that are different from those seen during training. To facilitate this, we generate a dataset of 150k distinct training instructions covering basic rearrangement concepts, from single-object interactions to finding and moving two objects. The train instructions include several phrasings of the same rearrangement concept and cover interacting with various objects and receptacles. For evaluation, we construct an evaluation dataset that tests the generalization capabilities with respect to different rearrangement concepts expressed with language (see Table 1 and Appendix B.5). All evaluation datasets are in unseen houses.

**Paraphrastic Robustness (PR):** The ultimate goal of training models capable of solving tasks from natural language instruction is to allow humans to easily provide instructions to embodied agents. However, humans exhibit high variability in how they describe a task (Schreitter & Krenn, 2014), so it is necessary that agents are robust to paraphrasing. *Paraphrasic Robustness* (PR) evaluates if the agent can produce the same behavior under a linguistic variation. Such variations include new ways of saying the instruction and referring to objects indirectly rather than by name. The underlying goal of these instructions are contained in the training dataset.

**Behavior Generalization (BG)**: The second category of datasets test *Behavior Generalization* (BG) in which the agent has to demonstrate a new type of behavior specified by the language instructions not present in the training dataset. This involves a new logical expression. For example, during training the agent was told how many objects to find. But BG includes the *Multiple Rearrangements* task, which requires an agent to find "all" of a particular object.

We believe these two axes of generalization have large coverage in terms of realistic semantic situations that a robot could encounter. PR and BG roughly align with generalization concepts in psychology; PR is similar to what is referred to as "stimulus generalization" in psychology, while BG can be thought of as a type of "response generalization" (Shepard, 1957), though in the latter we do not keep the linguistic instruction – the "stimulus" – fixed.

## 5 EXPERIMENTS

### 5.1 BASELINES

We compare LLaRP to the following baselines. For all methods in Language Rearrangement, we use the pre-trained VisualCortex model (VC1, Majumdar et al., 2023) a ViT backbone designed for egocentric visual tasks. We represent the visual observations using the ViT *[CLS]* token from VC1. We freeze the VC1 visual encoder module. All RL methods are trained with PPO. Further details for all methods are in Appendix C.

- **ZS-ChatGPT/ ZS-LLaMA**: Input the instruction to the LLM and zero-shot (ZS) plan a sequence of high-level actions. This policy is blind and plans based only on the language instruction. The prompt lists all the constraints (e.g., only one object can be picked at a time), possible actions, receptacles and objects, along with examples of successful behavior. We compare against an instruction-tuned LLaMA-65B (ZS-LLaMA) (Touvron et al., 2023) which generates a plan only

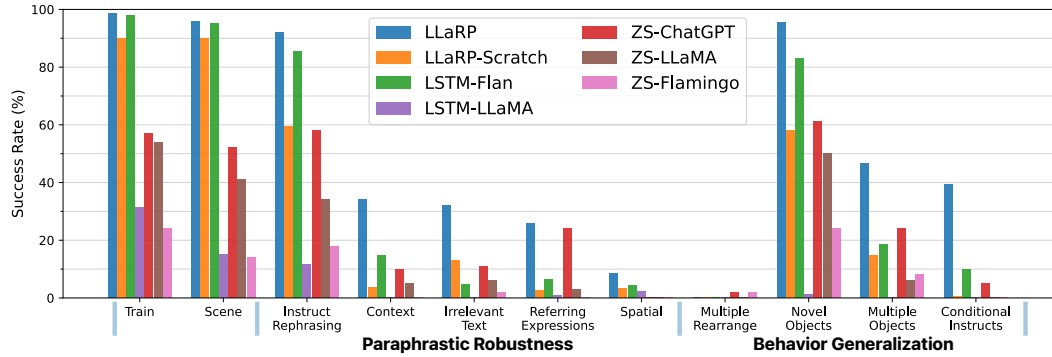

Figure 3: Success Rates across all evaluation tasks, aggregated in groups per Table 1. Across all unseen task groups, LLaRP generalizes better than the respective baselines, apart from multiple rearrangement, in which all models perform poorly. For exact numbers, see Table 5.

once at the beginning. We also compare against ChatGPT (ZS-ChatGPT) that performs multi-step reasoning to iteratively refine the plan based on proprioceptive feedback from the environment. For details about the ZS baselines, prompt, and environment feedback see Appendix C.4.

- **ZS-Flamingo**: Multimodal (text + image) version of the ZS-LLaMA baseline. Uses IDEFICS, an open source 80B parameter VLM model (Laurençon et al., 2023) based on Flamingo (Alayrac et al., 2022). Given a prompt similar to ZS-LLaMA, the image observation and the textual instruction, ZS-Flamingo outputs a single plan for the agent to follow.
- **LLaRP-Scratch**: Same architecture as LLaMA-7B but with 2 billion parameters. The entire transformer is trained from scratch and the pretrained visual encoder is frozen.
- **LSTM-Flan/ LSTM-LLaMA**: The instruction is encoded as a fixed-length vector that an LSTM takes as input along with the observation encoding. The action is predicted from the LSTM hidden state. LSTM-Flan uses Flan-T5 (Chung et al., 2022) to encode the instruction while LSTM-LLaMA uses LLaMA and a Perciever Resampler network (Jaegle et al., 2021).

## 5.2 EMPIRICAL ANALYSIS

In this section, we analyze the empirical performance of LLaRP and the baselines on Language Rearrangement. We show LLaRP has better zero-shot generalization capabilities than the baselines on most of the unseen Language Rearrangement evaluation datasets. LLaRP also learns efficiently, learning faster during training and comparing favorably to training with expert demonstrations.

**LLaRP improves generalization across all generalization types.** We report aggregate success rates across all generalization datasets in Table 2. We report the mean and standard deviation across 3 random seeds for all RL-based methods. We see that LLaRP is almost 1.7x better than the next best performing baseline, 42% vs. 25%. This trend is true for both Paraphrastic and Behavior generalizations, which shows that the use of a LLM allows for a model that can better understand natural language and execute novel tasks. Althought this is expected for paraphrastic robustnes as LLMs are known for their language understanding capabilities, it is somewhat surprising to see that the LLM helps even for novel behaviors, achieving 45% vs. 28% from LSTM-Flan.

Results across individual generalization test sets are shown in Figure 3. LLaRP displays superior generalization capabilities across all settings. Most learned methods achieve near 100% success rates on the train split and almost 100% when one varies the scene, but uses the same instructions as in train.

| Method | Total | Paraphrastic Robustness | Behavior Generalization |
|---|---|---|---|
| LLaRP | **42** ± **2** | **38** ± **1** | **45** ± **3** |
| LLaRP-Scratch | 17 ± 4 | 16 ± 3 | 18 ± 5 |
| LSTM-Flan | 25 ± 1 | 23 ± 1 | 28 ± 1 |
| LSTM-LLaMA | 2 ± 1 | 3 ± 2 | 0 ± 0 |
| ZS-ChatGPT | 22 | 21 | 23 |
| ZS-LLaMA | 12 | 10 | 14 |
| ZS-Flamingo | 6 | 4 | 8 |

Table 2: Combined zero-shot success rate. LLaRP outperforms all baselines in all categories.

LLaRP performs 7% and 12% better when we rephrase instructions or talk about novel objects compared to the next best performing model. However, the boost LLaRP gets from using a LLM is substantial for more complex novel tasks, e. g. when we use a context, have a conditional statement, multiple chained rearrangements, etc.

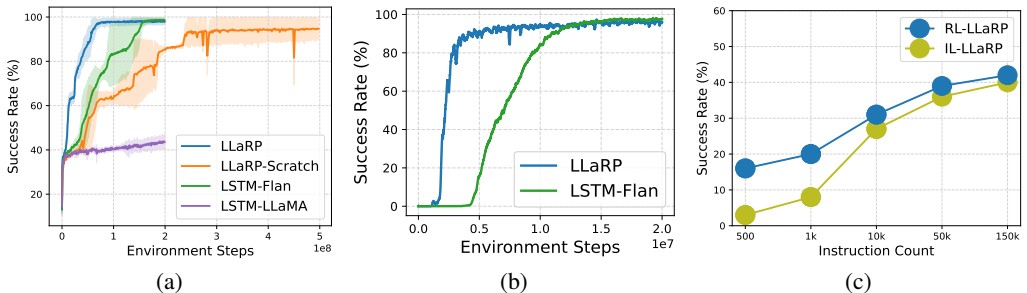

Figure 4: Success rates (SR) for different stages of training (left and middle) or number of episodes / demonstrations (right). (a) SR vs number of steps during learning. (b) SR vs number of steps during continual learning on "Multiple Rearrangments" tasks. (c) LLaRP with RL vs LLaRP with IL with the same number of episodes / demonstrations. See text for further discussion.

The second best performing model is LSTM-Flan, which like LLaRP uses a billion parameter language model (Flan-T5). However, LSTM-Flan uses it to comprehend instructions through encoding the instruction to a fixed size vector, rather than using the LLM as a decoder like LLaRP. LSTM-Flan generalizes in some scenarios such as to new phrasings and objects, indicating that the LSTM policy learned to interpret these aspects of the FLAN embedding. Whereas unlike LLaRP, LSTM-Flan performs worse on Behavior Generalization than Paraphrasic Robustness, indicating that using the LLM directly as a decoder is important for generalization to new behaviors. We also compare to LSTM-LLaMA which uses the same LLaMA hidden state activations as LLaRP. However, it performs generally worse, perhaps because LLaMA was not trained as an encoder model.

The zero-shot application of LLM, ZS-ChatGPT, requires no training and cannot receive images as inputs. Nevertheless, it achieves better-than-random performance across many of the evaluation splits, which demonstrates that the LLM contains relevant information for embodied tasks.

**LLMs lead to faster learning convergence.** As shown in Figure 4a, LLaRP is the most sample efficient model, converging in around 50M-100M steps before LSTM-Flan, despite being a substantially larger model. Further, LLaRP-Scratch takes 10x more environment samples to converge than LLaRP (50M versus 500M) despite both having the same architecture, showing that pre-trained LLMs are a good fit for the problem space.

Note that only a few prior works have used transformers for online RL (Parisotto et al., 2020). In Appendix D.1, we further explain the training architecture and settings that were critical to stably train LLaRP and baseline transformer policy training with PPO.

**LLMs leads to faster *continual* learning convergence.** To evaluate the continual learning efficiency, we continue learning the model on downstream tasks beyond the training tasks and analyze the training convergence. Specifically, we take the "Multiple Rearrangements" dataset, generate 10K episodes, and train until convergence. The results, shown in Figure 4b, show that LLaRP is 3x more efficient than LSTM-Flan, achieving near perfect performance in 500k vs 1.5M environment steps. Hence, LLMs can lead to faster learning of additional tasks.

**LLaRP with RL performs better and utilizes supervision more efficiently than IL.** A common paradigm of endowing LLMs with novel capabilities is is Imitation Learning (IL) with decision making data. More recently, works have endowed LLMswith decision making capabilities in embodied settings (Driess et al., 2023; Brohan et al., 2023).

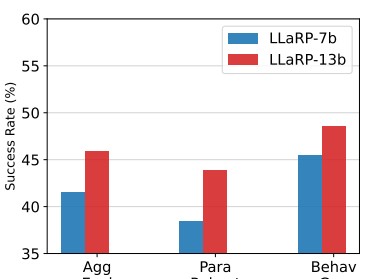

Figure 5: Average success rates across our evaluation datasets for LLaRP scaled from Llama-7b to Llama-13b.

Hence, a natural question is to compare LLaRP trained with RL vs IL. IL uses demonstrations generated by a fully trained LLaRP policy. For each data point, we use the same number of episodes for RL as demonstrations for IL, both denoted by instruction count, and train until convergence.

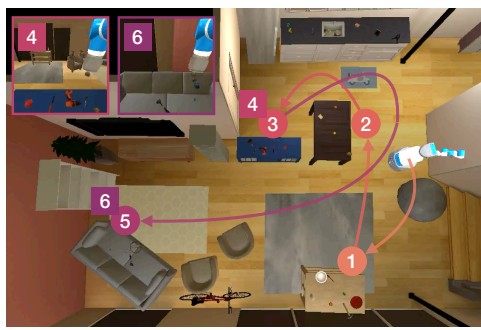

Figure 6: Success trajectory in the *Context* dataset. Arrows indicate navigation to receptacles. Right shows actions selected by LLaRP. Upper-left show egocentric observations for some steps.

In Figure 4c, we see that for any number of episodes (or demonstrations) RL outperforms IL, with a larger margin in the low regime of data. This shows that in RL settings, an LLM-based LLaRP is able to collect episodes that are useful for policy improvement. For RL we merely need a reward definition, while in IL settings full demonstration trajectories are required, thus making RL less costly than IL for each instruction count.

**Larger LLMs lead to better results.** In Figure 5, we show the effect of scaling the size of the LLM in LLaRP. We compare using LLaMA-13B and LLaMA-7B in LLaRP. We find that LLaMA-13B gives a 4% boost in total evaluation performance. This indicates that larger, more capable LLMs, can translate to more capable embodied reasoning.

**Qualitative Result.** In Fig. 6, we show a success example of LLaRP on the *Context* dataset. The agent explores for the first 3 actions. Then it finds the screwdriver implied by the phrase "fix a loose screw" in the instruction. It then successfully brings that screwdriver to the couch. For more qualitative examples in the other datasets, see Appendix E and videos at https://llm-rl.github.io.

**LLMs boost performance with tasks beyond Language Rearrangement.** We evaluate LLaRP across Atari 2600 games using the Arcade Learning Environment (Bellemare et al., 2013), configured following the recommendations of (Machado et al., 2018). We train LLaRP and LLaRP-Scratch as in Habitat, with the exception of using a fully-trainable ResNet-18 visual encoder, as Atari is visually distinct from the training data used for VC-1. To take advantage of the diversity in available Atari environments, we train LLaRP and LLaRP-scratch individually on each game of 55 Atari games for 100M environment steps, and report human-normalized scores (Mnih et al., 2013).

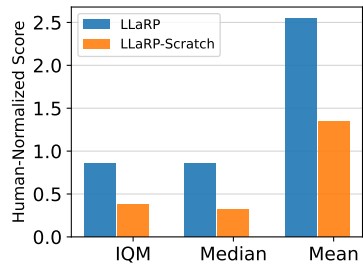

Figure 7: LLaRP and LLaRP-Scratch performance over 55 Atari games (two seeds per game).

We find that LLaRP outperforms LLaRP-Scratch by a large margin, with LLaRP achieving a higher average score on 43 out of 55. When aggregated, LLaRP's mean, median, and interquartile mean (Agarwal et al., 2021) performance are between 3x and 4x higher than LLaRP-Scratch (see Figure 7). Additional details about our Atari experiments can be found in Appendix C.6.

**Further Experiments.** Additional analyses in Appendix D show the impact of batch size and LLM weights for RL with transformers, unfreezing LLM weights for LSTM-Flan, LLaRP efficiency, and that our findings hold in a harder setting with no invalid actions and a termination action.

## 6 CONCLUSION

We introduce LLaRP, a scheme for employing pretrained LLMs on embodied tasks with reinforcement learning. To aid in our research, we introduce a dataset of rearrangement tasks (consisting of 150k training instructions and 10 challenging evaluation datasets). LLaRP outperforms non-pretrained transformer- and LSTM-based models on both sample efficiency and generalization. Limitations to be addressed in the future include the significantly larger size of LLMs than typical RL models. Training the action decoder module may also hinder the policy's ability to leverage the world knowledge from the LLM. Future work can explore how LLaRP can directly interact with the environment via the language head of the LLM, without the need for an action decoder module.

## 7 ACKNOWLEDGEMENTS

The authors would like to thank Eduord Grave, Mathias Muller, Josh Susskind, Barry Theobald, Yizhe Zhang for valuable comments and discussion.

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

## A CONTRIBUTIONS

**Andrew Szot** co-initiated the project, design both the benchmark and co-designed and implemented the algorithmic solution, conducted all RL and IL experiments on Habitat and their analysis, and co-wrote the paper.

**Max Schwarzer** ran experiments on Atari and co-wrote the paper.

**Harsh Agrawal** implemented, improved, and analyzed the zero-shot baselines, worked on dataset and evaluation methodology, and co-wrote the paper,

**Bogdan Mazoure** help with debugging training setup and co-wrote the paper.

**Rin Metcalf Susa** investigated non-LLaMA LLMs and edited the paper.

**Walter Talbott** ran model probing analysis and contributed to discussions.

**Natalie Mackraz** worked on RL policy text input and provided feedback for the paper.

**Devon Hjelm** advised on experiment setup, helped shape up the overall story of the work, and co-wrote the paper.

**Alexander Toshev** co-initiated the project, managed and advised throughout the project, co-designed the algorithmic solution, and co-wrote the paper.

## B LANGUAGE REARRANGEMENT DETAILS

### B.1 PLANNING DOMAIN DEFINITION LANGUAGE (PDDL) DETAILS

Language Rearrangement is represented with a **Planning Domain Definition Language (PDDL)** specification (Aeronautiques et al., 1998). Specifically, each episode is linked to a PDDL problem specification. This specification consists of the following components:

- **Entity types**: Each entity in the scene is associated with an entity type. This entity type is used to determine which predicates are applicable. The type system is hierarchical, and derived types also apply to higher level types. We define core types such as *pickable_object* and *receptacle*. We automatically derive types from the object dataset based on object high-level categories such as *fruit* and object classes such as *apple*.
- **Entities**: For each object and receptacle in the scene, the PDDL associates it with a symbolic *entity*. Each entity has an associated entity type. The entities are automatically populated for each rearrangement episode based on the loaded objects and receptacles.
- **Predicates**: These are binary expressions that are evaluated based on the underlying simulator state. For example, *on_top(X : pickable_object, Y : receptacle)* corresponds to if object *X* is on top of receptacle *Y*.

Between episodes, the only component that changes is the "**Entities**" category since each episode will have different objects. The same entity types and predicates apply between episodes.

A **predicate expression** refers to a boolean expression stated in first order logic involving the PDDL components. Predicate expressions involve the predicates, entities, entity types, logical connectives ("and", "or", "not") and quantifier symbols ("for all", "exists"). For example, the predicate expression $\exists x : is\_type (x, "apple"), on\_top(x, "table")$ will be evaluated to true, when any apple is placed on top of the table, and false otherwise. As described in the next section, we use predicate expressions to describe success criteria for instructions.

## B.2    INSTRUCTION GENERATION DETAILS

We create a scalable pipeline for generating a large number of satisfiable, plausible instructions from a small number of *instruction templates*. Each instruction is associated with a predicate expression defining the success criteria for that instruction. These components are described in detail below.

**Instruction Template**: This refers to a particular outcome in the environment and corresponding ways of expressing this outcome in language where nouns are replaced with placeholder variables. Specifically, an instruction template consists of:

- **Template Goal Condition**: This describes the desired outcome for the instruction template represented as an *ungrounded predicate expression* (predicate expressions are described in Appendix B.1). It is *ungrounded* because there are placeholder variables for entities rather than actual entities in the scene to refer to multiple outcomes depending on what entities are substituted into the placeholders. For example, for the ungrounded desired outcome of an object going on a receptacle, the goal condition would be the predicate expression $\exists x : is\_type (x, "OBJECT"), on\_top(x, "RECEPTACLE")$, where *OBJECT* and *RECEPTACLE* are placeholders.
- **Placeholder Constraints**: The template goal condition used a set of placeholder variables. But in the predicate expression $\exists x : is\_type (x, "OBJECT"), on\_top(x, "RECEPTACLE")$ we need *OBJECT* to be a pickable entity type and *RECEPTACLE* to be a receptacle entity type. The instruction template thus contains constraints on the entities that can be sampled for the template goal condition.
- **Instruction Language Templates**: Each instruction template has a set of $N$ ways of expressing the template goal condition in *ungrounded natural language*. It is *ungrounded* because the language will contain placeholder variables that will be later substituted for entities in the scene. For example, "Move an 'OBJECT' to the 'RECEPTACLE'" is a language template for the template goal condition $\exists x : is\_type (x, "OBJECT"), on\_top(x, "RECEPTACLE")$. We set $N$ to 11, meaning each template goal has 11 different language templates, each expressing the same template goal condition with different language.

We define a set of instruction templates for each of the datasets from Section 4.2. We show examples of the language templates from these instruction templates for all the datasets in Table 7.

The next step of our pipeline uses these instruction templates to generate a dataset of rearrangement *episodes*. When generating a particular dataset, we start by randomly sampling a template from that dataset. We then substitute random entities into the template placeholders, taking into account the placeholder constraints. We then randomly sample a scene that is compatible from the associated scene set. A scene is compatible with an instruction if all the receptacles the instruction refers to are present in the scene. We then populate the scene with objects. We constrain the object sampling for the template. We include all template substituted entities and constraints in the object sampling process. Thus an episode consists of:

- **Scene**: The empty scene specifying the house floor plan.
- **Entity Locations**: The transformations of all receptacles and objects.
- **(Grounded) Goal Condition**: A predicate expression describing the desired outcome without any placeholder variables.
- **(Grounded) Instruction Language**: An instruction in natural language specifying the goal condition without any placeholder variables.

Note that the robot starting transformation is not included in the above, and is instead randomly generated at the start of every episode.

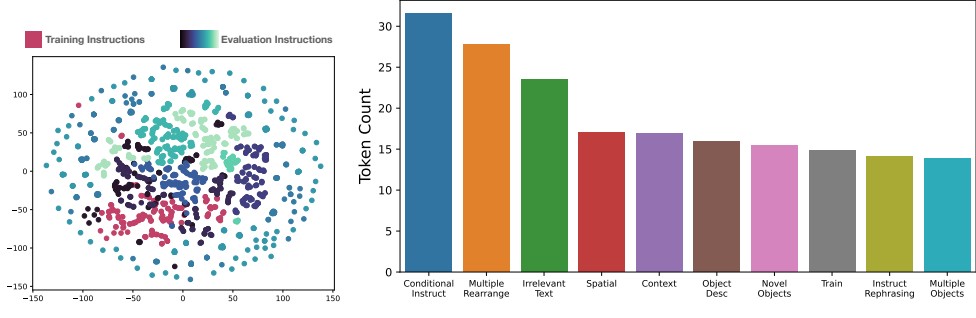

(a) Instruction embedding T-SNE.  (b) Avg Tokens per Instruct

Figure 8: Left: Visualization of the evaluation and training dataset instructions. We embed the instruction using Flan. Only the instructions depicted in red are seen during training. Right: The average number of language tokens per instruction in each dataset.

We then check that each generated episode is solvable. Random object placement and physics simulation can result in objects falling and other instabilities. We check the episode is solvable by an oracle planner. We run a STRIPS based planner Fikes & Nilsson (1971) that operates from the PDDL problem instantiated by the episode. If the planner times out or the solution is longer than 32 high-level steps (a high-level corresponds to a single skill, defined in Appendix B.3), we consider the episode unsolvable and remove it from the training set.

## B.3 Skill Details

We train the policies in an abstracted, high-level action space. Each high-level action corresponds to a particular skill invocation, for example, " *pick(apple)*". We consider the following skills:

- **Navigation**: Parameterized by the name of the receptacle to navigate to. So long as the receptacle is present in the scene, this skill is always valid
- **Pick**: Parameterized by the name of the object to pick. Only valid if the robot is close to the object, not holding another object, and the object is not inside a closed receptacle.
- **Place**: Parameterized by the name of the receptacle to place the object on. Only valid if the robot is close to the receptacle and is holding an object.
- **Open**: Parameterized by the name of the receptacle to open. Only valid if the receptacle is closed and the robot is close to the receptacle.
- **Close**: Parameterized by the name of the receptacle to close. Only valid if the receptacle is open and the robot is close to the receptacle.

The skill is only executed if it is valid in the current state. The action space consists of all possible skill parameterizations given all possible objects, giving 70 total skills meaning 70 actions for the policy to select from. Note that some of these skills may be applied to objects that are not present in the current scene, in which case selecting that skill would be an invalid action. Furthermore, the actions are fixed at every step, and thus a majority of the actions will be invalid at a given step. We treat invalid actions as no-ops, but we compare to where the robot is not allowed to take invalid actions in Appendix D.3. If the action is valid, we execute it and instantaneously transform the simulator state based on the effect of the skill. For example, if the skill "pick(apple)" is selected and the robot is near the apple, not holding anything, and the apple is not in a receptacle, then the apple will teleport to the robot's gripper.

## B.4 Additional Task Details

Language Rearrangement is simulated in Habitat 2.0 (Szot et al., 2021). The task is simulated with kinematic dynamics. As described in Appendix B.3, after the agent selects a valid skill, the simulator state is instantaneously transformed to the skill post-condition. Only valid skills are executed. Invalid skills that result in collisions, impossible interactions (such as the agent grabbing an object out of reach) are counted as invalid actions.

| Hyperparameter | LLaRP | LLaRP-Scratch | LSTM-Flan | LSTM-LLaMA |
|---:|:---:|:---:|:---:|:---:|
| LR | $3e^{-4}$ | $3e^{-4}$ | $3e^{-4}$ | $3e^{-4}$ |
| Optimizer | AdamW | AdamW | Adam | Adam |
| Number of Mini Batches | 6 | 6 | 4 | 4 |
| Environments Per GPU | 18 | 18 | 18 | 18 |
| Entropy Coefficient | 0.01 | 0.01 | 0.01 | 0.01 |
| Value Loss Coefficient | 0.5 | 0.5 | 0.5 | 0.5 |
| Number of Rollout Steps | 32 | 32 | 32 | 32 |
| Number of PPO Epochs | 1 | 1 | 2 | 2 |
| Batch Size Per Update | 768 | 768 | 1152 | 1152 |

Table 3: Hyperparameters for all RL methods.

The episode is considered successful if the goal condition evaluates to true. The episode is a failure if the agent doesn't achieve success in under 32 high-level policy steps. We consider requiring the agent to call a separate stop action in Appendix D.3. We use the objects from the YCB (Calli et al., 2015) and ReplicaCAD (Szot et al., 2021) object datasets. The object categories used are: "ball, clamp, hammer, screwdriver, padlock, scissors, block, drill, spatula, knife, spoon, plate, sponge, cleanser, plum, pear, peach, apple, lemon, can, box, banana, strawberry, lego, rubriks cube, book, bowl, cup, fork". We hold out "mug, orange, lid, toy airplane, wrench" for the *Novel Objects* evaluation split.

The Language Rearrangement reward function consists of a sparse reward for completing the task, subgoal rewards for completing individual parts of the task, and a slack penalty for completing the task faster. The reward at step $t$ is:

$$r_t = 10 \cdot \mathbb{1}_{\text{success}} + 5 \cdot \mathbb{1}_{\text{subgoal}} - 0.1 \cdot \mathbb{1}_{\text{invalid}} - 0.01$$

Where $\mathbb{1}_{\text{success}}$ indicates if the PDDL goal expression is evaluated as true, meaning the episode was successfully solved. $\mathbb{1}_{\text{invalid}}$ indicates if the agent called an invalid action at the current step. $\mathbb{1}_{\text{subgoal}}$ indicates if the agent achieved any subgoal necessary to achieve the overall goal. For example, for the instruction "Find an apple and put it away in the fridge", the agent needs to first pick up the apple and potentially open the fridge. When running the STRIPS planner in the episode validation process described in Appendix B.2, we compute the optimal action sequence and extract subgoals from this action sequence. We then reward the agent for reaching any of these high-level subgoals. Note we cannot directly imitate the optimal action sequence because it is computed with oracle state information and is an "impossibly good" expert (Walsman et al., 2022).

### B.5 Additional Dataset Details

Further details on the Paraphrasic Robustness datasets:

- **Instruction Rephrasing**: The same underlying instruction goals, but stated in a different way. For example, a rephrasing of the training instruction. The order that nouns appear in the instruction is permuted and synonyms for verbs are substituted.
- **Referring Expressions**: Refer to objects by their visual appearance rather than directly as their entity name. For example, an "apple" is referred to by its visual appearance as a "round red fruit". During training, objects are only referred to by a *single* name. The referring expression was never seen during training.
- **Spatial Relationships**: This refers to receptacles indirectly by their location relative to other receptacles. All receptacles are positioned against the walls, so there is no spatial ambiguity depending on the agent viewpoint. There are no spatial concepts in the training data.
- **Context**: Describe a situation where a particular object fits the context.
- **Irrelevant Instruction Text**: Instructions that include irrelevant context.

Further details on the Behavior Generalization datasets:

- **Multiple Rearrangements**: Generalize to rearranging 3 objects when the agent only rearranges 2 objects during training.

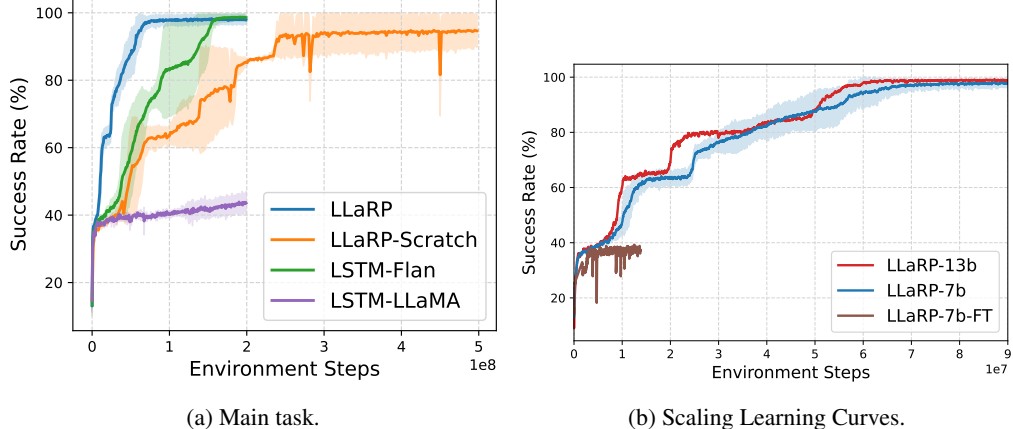

(a) Main task.                                    (b) Scaling Learning Curves.

Figure 9: Left: Learning curves for all baselines in the main Language Rearrangement task from Figure 3. Note that we train all methods for 200M steps except for LLaRP-Scratch, as we observed that it had not converged and was still making learning progress at that point. Right: Learning curves for the scaling results from Figure 5.

- **Novel Objects**: New entity, instruction pairs. We holdout particular objects from single object pick and place instructions during training. This split evaluates on the single object pick and place instructions with this holdout object.
- **Multiple Objects** : Different rearrangement concepts. The agent has never seen the concept of "for all" in the training dataset. This requires the agent to rearrange a variable number of objects depending on how many entities are in the scene. The agent may need to rearrange 1 to 2 objects in the scene since the number of each object is randomized during scene generation. Therefore, the agent must search to detect that all the objects belonging to the specified type have been moved.
- **Conditional Instructions** : Adjust behavior based on if the conditional statement is true. In Language Rearrangement we consider changing the behavior based on if the fridge is open or closed. We split this dataset into the fridge being open half of the time. The agent has to find and move one of two objects depending on if the fridge is closed or not. For the object that doesn't need to be moved, the agent must also not displace this object.

### B.6 DATASET ANALYSIS

In this section, we study how the evaluation datasets differ from the training dataset and each other in more detail. We note the high-level differences below:

- Each dataset consists of different instruction templates (instruction templates are described in Appendix B.2). Different instruction templates mean the underlying goal conditions and language instructions will be different.
- All non-train datasets are on unseen scenes. These unseen scenes are from the evaluation split from ReplicaCAD (Szot et al., 2021). The location of the kitchen and furniture differs from training.
- Objects are placed in new locations. In every episode, objects start in a new position. Thus we always evaluate on unseen object positions, even for the *train* split.

We qualitatively analyze the differences between datasets by visualizing language embeddings of the instructions between the different instructions in Figure 8a. Specifically, we take 100 random instructions from each dataset and embed them using the Flan-T5-XL model (Chung et al., 2022). These embeddings are then visualized as a T-SNE plot. The training instructions, visualized as red points, only have minimal overlap with the evaluation instructions.

Additionally, we display the average number of tokens per instruction (see Figure 8b). From here we can see that our evaluation splits have variability at the token-count level, which in the largest case (for conditional instruction datasets) have approximately double the length of prompts than the median task.

Diving into this analysis further, in Figure 14, we display the per-word frequencies in the instructions. As expected, we see a long-tailed distribution across the board, favoring receptacles and relative indicators for nouns (e.g. counter, table, left) and instructions for verbs (e.g. move, place, and swap). It also is no surprise that when looking at all words in the combined plot, articles are the most common. Overall, this shows the diverse language across our instructions, and aids in verifying that our instruction generation is acting as expected.

### B.7 COMPARING LANGUAGE REARRANGEMENT TO OTHER BENCHMARKS

In Appendix B.7, we compare Language Rearrangement to related embodied AI and interactive instruction following benchmarks. The primary difference from Language Rearrangement and other benchmarks, is that Language Rearrangement has more instructions (almost 19x more than the closest work ALFRED (Shridhar et al., 2020)), generated by more instruction templates (8x more than CALVIN Mees et al. (2022)). Also unique about Language Rearrangement is that it has dense rewards for the language specified tasks (the dense reward is described in Appendix B.4). All other tasks with language instructions only define sparse success-based rewards. Since Language Rearrangement is implemented with Habitat 2.0 (Szot et al., 2021), it also inherits the same fast simulation speeds.

While Language Rearrangement supports low-level skill execution since it is built on Habitat 2.0 which supports dynamic physics, we do not evaluate with dynamically simulated low-level skills and instead kinematically update the simulator state to the skill post condition. This is similar to the high-level interaction space in ALFRED (Shridhar et al., 2020). Other tasks focus on low-level control, and thus simulate the full robot control, but are constrained to the more limited setting of a robot arm fixed on a tabletop table top (Mees et al., 2022; Gong et al., 2023; Shridhar et al., 2022; Zeng et al., 2021).

| Benchmark | # Instructions (Tasks) | # Instruction Templates | Observation Type | Reward Type | Generalization Type | Sim Speed steps/second | # Scenes | # Objects |
|---|---|---|---|---|---|---|---|---|
| **Language Rearrangement (Ours)** | 151,000 | 282 | Visual | Dense Reward | Unseen Instructions [10 datasets from Tab. 1], Unseen Scenes | 1400 | 105 | 82 |
| ALFRED Shridhar et al. (2020) | 8,055 | 7 | Visual | Sparse Success | Random Split | NA | 120 | 84 |
| CALVIN Mees et al. (2022) | 400 | 35 | Visual | Sparse Success | Unseen Instructions | NA | 1 | 1 |
| ARNOLD Gong et al. (2023) | 32 | 8 | Visual | Sparse Success | Random Split | 200-400 | 20 | 40 |
| CLIPort (Ravens) Shridhar et al. (2022); Zeng et al. (2021) | 10 (+Procedural) | 10 | Visual | Sparse Success | Unseen Objects [colors, shapes, types] | NA | 1 | 56 |
| BabyAI Chevalier-Boisvert et al. (2019); Carta et al. (2023) | Procedural | Procedural† | Text | Sparse Success | Unseen Instructions [compositions, objects, synonyms, dialects] | 3000 | NA | 4 |
| Habitat Rearrangement Szot et al. (2022) | NA | NA | Visual | Dense Reward | Unseen Scenes | 1400 | 105 | 20 |
| Behavior1k Li et al. (2023a) | NA | NA | Visual | Sparse Success | NA | 60 | 50 | 5215 |
| TDW Gan et al. (2020) | NA | NA | Visual | Sparse Success | Unseen Scenes | 15 | 5-168 | 112 |
| ProcTHOR Deitke et al. (2022) | NA | NA | Visual | Dense Reward | Unseen Scenes | 90-180 | 10k | 118 |

Table 4: Comparing Language Rearrangement to similar benchmarks. "NA" under the simulator speed indicates the benchmark is used for offline learning and not for RL where simulation speed is a concern. "NA" under instructions indicates the task does not provide language instructions and instead features a task specification such as a target object name to navigate to or rearrange. "NA" under generalization type means the task does not evaluate trained policies in unseen task variations, and instead focuses on within train task distribution performance. † BabyAI uses an instruction language grammar instead of instruction templates. This grammar has 4 clauses, 10 attributes (locations and colors), and 4 nouns.

## C METHOD DETAILS

In this section, we describe the architectures, training procedure of LLaRP, and baselines in more detail. Unless specified otherwise, every method is trained using a full node of 8 A100-80GB GPUs

| | Total | Aggregated Behavior Generalization | Paraphrastic Robustness | Train | Scene | Instruct Rephrasing | Novel Objects | Per Dataset Breakdown Multiple Rearrange | Referring Expressions | Context | Irrelevant Text | Multiple Objects | Spatial | Conditional Instructs |
|---|---|---|---|---|---|---|---|---|---|---|---|---|---|---|
| LLaRP | 42±2 | 45±3 | 38±1 | 99±1 | 96±4 | 92±2 | 95±4 | 47±5 | 26±2 | 34±2 | 32±2 | 0±1 | 8±1 | 39±3 |
| LLaRP-Scratch | 17±4 | 18±5 | 16±3 | 90±9 | 90±9 | 59±13 | 58±16 | 15±6 | 3±1 | 4±3 | 13±4 | 0±0 | 3±3 | 1±1 |
| LSTM-Flan | 25±1 | 28±1 | 23±1 | 98±1 | 95±8 | 85±2 | 83±3 | 19±4 | 6±1 | 15±3 | 5±2 | 0±0 | 4±4 | 10±6 |
| LSTM-LLaMA | 2±1 | 0±0 | 3±2 | 31±2 | 15±2 | 12±3 | 1±1 | 0±1 | 1±2 | 0±1 | 0±1 | 0±0 | 2±4 | 0±0 |
| ZS-ChatGPT | 22 | 23 | 21 | 57 | 52 | 58 | 61 | 24 | 24 | 10 | 11 | 2 | 0 | 5 |
| ZS-LLaMA | 12 | 14 | 10 | 54 | 41 | 34 | 50 | 6 | 3 | 5 | 6 | 0 | 0 | 0 |
| ZS-Flamingo | 6 | 8 | 4 | 24 | 14 | 18 | 24 | 8 | 0 | 0 | 2 | 2 | 0 | 0 |
| LLaRP-FT | 1 | 0 | 1 | 30 | 13 | 5 | 0 | 0 | 0 | 0 | 0 | 0 | 0 | 0 |
| LLaRP-7b | 42±2 | 45±3 | 38±1 | 99±1 | 96±4 | 92±2 | 95±4 | 47±5 | 26±2 | 34±2 | 32±2 | 0±1 | 8±1 | 39±3 |
| LLaRP-13b | 46 | 48 | 44 | 98 | 100 | 95 | 98 | 51 | 31 | 41 | 37 | 0 | 15 | 45 |
| LLaRP (HL) | 42±2 | 45±3 | 38±1 | 99±1 | 96±4 | 92±2 | 95±4 | 47±5 | 26±2 | 34±2 | 32±2 | 0±1 | 8±1 | 39±3 |
| LSTM-Flan (HL) | 25±1 | 28±1 | 23±1 | 98±1 | 95±8 | 85±2 | 83±3 | 19±4 | 6±1 | 15±3 | 5±2 | 0±0 | 4±4 | 10±6 |
| LLaRP (Harder) | 28 | 27 | 28 | 56 | 61 | 62 | 56 | 31 | 23 | 32 | 24 | 0 | 1 | 20 |
| LSTM-Flan (Harder) | 12 | 14 | 11 | 57 | 52 | 50 | 43 | 11 | 3 | 0 | 2 | 0 | 0 | 0 |

Table 5: Zero-shot results on Language Rearrangement for all baselines and settings. This includes the numbers from the bar plots in Figure 3, Figure 5 and Figure 12a. All numbers except for non-RL methods, LLaRP-FT, LLaRP-13b, and the harder task setting are mean and standard deviation over 3 random seeds.

and 96 Intel(R) Xeon(R) CPUs @ 2.20GHz. The base models are able to fit on a single GPU and we use data parallelism. Each GPU runs 1 policy and a set of $N$ environment workers. During the PPO rollout phase, the policy acts in parallel in all $N$ environments. Each GPU then computes the PPO update and synchronizes gradients. We use DD-PPO (Wijmans et al., 2019) to handle straggler environment workers and speed up synchronization between GPUs.

Hyperparameters for all reinforcement learning based methods are summarized in Table 3. Next, we detail specific method architecture choices.

## C.1 LLaRP Details

In general, the visual encoder can produce $M$ embeddings per observation $o_t$, which consists of a high-dimensional visual component (the robot's RGB camera) and a low-dimensional state component (the robot joint angles). The visual component is projected to a set of $M - 1$ tokens. For example, using a ViT (Dosovitskiy et al., 2020) for $E_\phi^V$ produces an embedding per image patch which are projected to $M - 1$ embeddings using a Perceiver Resampler Jaegle et al. (2021) network. The state components are concatenated and projected with an MLP to produce another embedding, giving $M$ total embeddings. Since there are now $M$ tokens per observation $o_t$, to produce a distribution over actions, we skip the first $k$ tokens corresponding to instruction tokens, and sub-sample every $M^{\text{th}}$ hidden output to extract just one action per time step.

However, in this work we just use the visual RGB observation and set $M = 1$. All methods use the frozen VC1 visual encoder whose weights are represented in bfloat16. We then take the *[CLS]* token of the VC1 encoder for the image observation. We then input this embedding to a linear projection layer that produces an embedding the same dimension as the LLM text tokens. For the action output module, we use a 2-layer MLP with ReLU activations, LayerNorm, and a hidden dimension size of 512.

By default, we use the base LLaMA-7B V1 (Touvron et al., 2023) for the LLM in LLaRP. We convert the LLaMA weights to bfloat16. The observation encoder and action output modules represent their weights in float32. The context window in Language Rearrangement is the maximum episode horizon of 32 steps. We compute the attention mask during training so the transformer only attends to inputs from the current episode. Despite running such a large policy with RL, we find that total training throughput is 700-800 steps-per-second on a full compute node of 8 GPUs. This timing includes policy inference for data collection, policy updates, and environment stepping including rendering and physics. The biggest bottleneck during training is policy inference.

## C.2 LLaRP-Scratch Details

For the transformer network, we use the same architecture and details as LLaMA. We restrict the transformer network size to around 2 billion parameters. As with LLaRP, we represent the transformer decoder network with bfloat16 data type. We update all parameters. Since the policy is smaller than regular LLaRP, the training throughput is even faster at 800-900 steps-per-second on a full compute node, despite updating all 2 billion policy parameters.

## C.3 LSTM-FLAN/LLAMA DETAILS

For LSTM-Flan, we only use the Flan-T5-XL encoder, and remove the decoder. The encoder is used to summarize the instruction. Specifically, we take the hidden activation of the final token as the instruction representation. We fine tune the weights of the Flan encoder which we show is necessary for good training performance in Appendix D.2. We represent the Flan weights in bfloat16. We represent the image as the *[CLS]* token from VC1 and process this embedding with a linear layer before concatenating it with the language representation and inputting it to the LSTM.

For LSTM-LLaMA, we use LLaMA-7B. The LLaMA weights are frozen and in bfloat16 format. For the instruction, we summarize the hidden outputs in a single instruction representation using a Perciever network.

## C.4 ZERO-SHOT BASELINE DETAILS

ZS-LLaMA uses a LLaMA-65B V1 (Touvron et al., 2023) model that was instruction tuned on ORCA style data (Mukherjee et al., 2023; Lian et al., 2023). Since, LLaMA-65B is a language-only model, this baseline is blind – it plans actions based only on the language instruction. To help the baseline reason about the available objects and actions, the prompt lists all receptacles and objects along with examples of successful behavior. For our multimodal baseline ZS-Flamingo, we use an open-source vision-and-language model IDEFICS (Laurençon et al., 2023) which is a reproduction of the closed-source Flamingo (Alayrac et al., 2022) model. IDEFICS uses LLaMA-65B V1 as its language model backbone. Its vision encoder is a vision transformer (ViT-H/14) trained using OpenCLIP (Radford et al., 2021) on the LAION-2B English subset of LAION-5B (Schuhmann et al., 2022) dataset. For both ZS-LLaMA and ZS-Flamingo, we use the following prompt:

```
You are a home robot assistant that can take actions in the house.
    Remember the following guidelines:
1. Your possible actions are: pick(object), place_on_recep(receptacle),
    navigate(receptacle), open_fridge(), close_fridge(),
    open_cabinet(cabinet), STOP.
2. Possible objects are: ball, clamp, hammer, screwdriver, padlock,
    scissors, block, drill, spatula, knife, spoon, plate, sponse,
    cleanser, plum, pear, peach, apple, lemon, can, box, banana,
    strawberry, lego, rubrik's cube, book, bowl, cup, mug, orange, lid,
    toy airplane, wrench.
3. You can only pick one object at a time.
4. If you place an object you must have previously picked it.
5. You must always place the object that you have picked.
6. To open a fridge, you have to navigate to the fridge.
7. To pick an object from the cabinet, you need to open it first.
8. place_on_recep() is not valid for cabinets like 'cabinet drawer 7' or
    'cabinet drawer 6'.
9. There are no more than 5 objects.
10. When exploring, select randomly from possible receptacles: [cabinet
    drawer 7, cabinet drawer 6, fridge, chair, black table, brown table,
    TV stand, sink, right counter, left counter]
11. When you are done output STOP.
12. If the instruction doesn't specify where the object is located, you
    should explore by navigating to a previously unvisited receptacle.
13. If the instruction asks to pick up more than one object, you should
    attempt multiple pick and place. Look for each object by visiting
    all the receptacles to find the objects mentioned in the instruction.
14. Don't get stuck in a loop by picking from and placing receptacle on
    the same receptacle.

To help you understand, here's are twos example:
# User: Instruction: Move the screwdriver from the left counter to the
    sofa.

# Assistant: navigate(left counter), pick(screwdriver), navigate(sofa),
    place_on_recep(sofa), STOP.
```

```
# User: Instruction: Find an apple and put it on the brown table.

# Assistant: navigate(fridge), open_fridge(), pick(apple),
    navigate(brown table), place_on_recep(brown table), STOP.
```

The ZS-ChatGPT baseline can perform multi-step reasoning to generate a plan. Unlike ZS-LLaMA, it is not limited to generating the whole plan in a single step. Instead, it can continuously refine the plan based on environment feedback. We use GPT-3.5-Turbo which is trained for chat applications for these experiments. The environment provides feedback in natural language consisting of agent's location (e.g. You are now at black table), failed action (e.g. Couldn't execute pick("apple"). Object wasn't found), and asks for the new plan. We update the prompt to also contain examples that require multiple steps of reasoning. The prompt is as follows:

```
#System: You are a home robot assistant that can take actions in the
    house. Remember the following guidelines:
1. Your possible actions are: pick(object), place_on_recep(receptacle),
    navigate(receptacle), open_fridge(), close_fridge(),
    open_cabinet(cabinet), STOP.
2. Possible objects are: ball, clamp, hammer, screwdriver, padlock,
    scissors, block, drill, spatula, knife, spoon, plate, sponse,
    cleanser, plum, pear, peach, apple, lemon, can, box, banana,
    strawberry, lego, rubrik's cube, book, bowl, cup, mug, orange, lid,
    toy airplane, wrench.
3. You can only pick one object at a time.
4. If you place an object you must have previously picked it.
5. You must always place the object that you have picked.
6. To open a fridge, you have to navigate to the fridge.
7. To pick an object from the cabinet, you need to open it first.
8. place_on_recep() is not valid for cabinets like 'cabinet drawer 7' or
    'cabinet drawer 6'.
9. There are no more than 5 objects.
10. When exploring, select randomly from possible receptacles: [cabinet
    drawer 7, cabinet drawer 6, fridge, chair, black table, brown table,
    TV stand, sink, right counter, left counter]
11. When you are done output STOP.
12. If the instruction doesn't specify where the object is located, you
    should explore by navigating to a previously unvisited receptacle.
13. If the instruction asks to pick up more than one object, you should
    attempt multiple pick and place. Look for each object by visiting
    all the receptacles to find the objects mentioned in the instruction.
14. Don't get stuck in a loop by picking from and placing receptacle on
    the same receptacle.

To help you understand, here's are twos example:
#User: Move the screwdriver from the left counter to the sofa.
#Assistant:
```
plan = [
  navigate("left counter"),
  pick("screwdriver"),
  navigate("sofa"),
  place_on_recep("sofa"),
  STOP
]
```

#User: Find an apple and a banana and put it on the left counter
#Assistant:
```
plan = [
  navigate("cabinet drawer 7"), # exploring an unvisited receptacle
  open_cabinet("cabinet drawer 7"), # opening the cabinet drawer 6 for
      the first time.
```

```
 pick("apple"),
 navigate("left counter"),
 place_on_recep("left counter"),
 STOP
]
```
#User: You are now at cabinet drawer. Couldn't execute pick("apple").
    Object wasn't found. Randomly select an unexplored receptacle in
    [cabinet drawer 7, cabinet drawer 6, fridge, chair, black table,
    brown table, TV stand, sink, right counter, left counter]
#Assistant:
```
plan = [
 navigate("fridge"), # exploring an unvisited receptacle
 open_fridge(),
 pick("apple"),
 navigate("left counter"),
 place_on_recep("left counter"),
 STOP
]
```
#User: You are now at fridge. Couldn't execute pick("apple"). Object
    wasn't found. Randomly select an unexplored receptacle in [cabinet
    drawer 7, cabinet drawer 6, fridge, chair, black table, brown table,
    TV stand, sink, right counter, left counter]
#Assistant:
```
plan = [
 navigate("brown table"), # exploring an unvisited receptacle
 pick("apple"),
 navigate("left counter"),
 place_on_recep("left counter"),
 STOP
]
```
#User: Found an apple.
#Assistant:
```
plan = [
 navigate("cabinet drawer 7"),
 open_cabinet("cabinet drawer 7")
 pick("banana"),
 navigate("left counter"),
 place_on_recep("left counter"),
 STOP
]
```
#User: You are now at cabinet drawer 7. Couldn't execute pick("banana").
    Object wasn't found. Randomly select an unexplored receptacle in
    [cabinet drawer 7, cabinet drawer 6, fridge, chair, black table,
    brown table, TV stand, sink, right counter, left counter]
#Assistant:
```
plan = [
 navigate("chair"),
 pick("banana"),
 navigate("left counter"),
 place_on_recep("left counter"),
 STOP
]
```
#User: Found the banana. Thanks!
""",
```

### C.5 MODIFICATIONS TO SCALE AND TRAIN LLaMA

For the LLaMA models larger than 7B parameters or with trainable parameters (unfrozen), training no longer fits on a single GPU. To train these LLaRP variants, we use model parallelism. Specifically, we distribute the model weights between 4 GPUs and scale training to 4 nodes (8 GPUs each) to match the batch size.

### C.6 MODIFICATIONS FOR ATARI

When training LLaRP on Atari games, we found that it was vital to allow the visual encoder to be fully trainable. Initial experiments with a frozen VC-1 visual encoder demonstrated some limited learning, but were unable to reliably solve Pong within 100M steps, our threshold for success. Even with this change, we observed frequent instabilities in our early experiments. Generally speaking, instabilities propagated from value learning to the actor over the course of 1-2 gradient steps. We were able to fix this by using a Huber loss for value learning, as well as by applying gradient norm clipping (with max gradient norm 0.5). We also apply reward clipping (Mnih et al., 2013) to stabilize our predicted values.

We also found that training at relatively large batch sizes – compared to language rearrangement – was beneficial. By default we used a per-device batch size of 64, with context length 32, leading to a total of 16,384 states being seen at once when training across 8 GPUs.

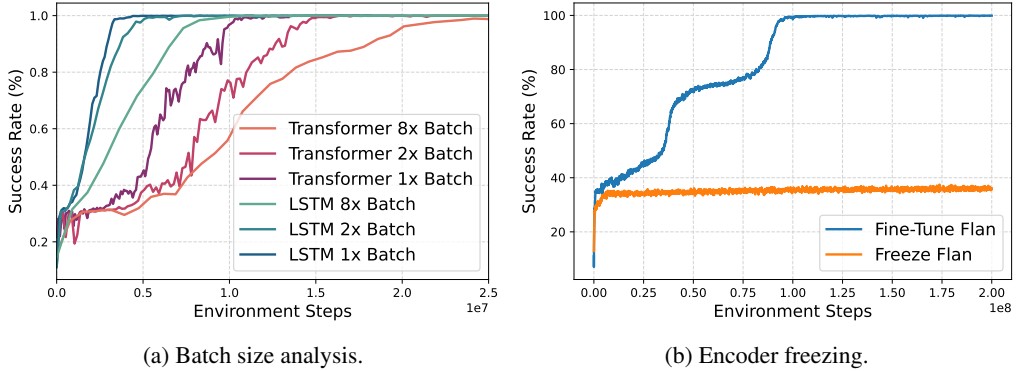

(a) Batch size analysis.          (b) Encoder freezing.

Figure 10: Left: Success rate of policies on the Train evaluation dataset. Right: Impact of freezing vs unfreezing the FLAN encoder in LSTM-Flan.

## D FURTHER RESULTS

In this section, we show further results and analysis on LLaRP and baselines.

### D.1 TRANSFORMER POLICY EXPERIMENTS

A primary challenge of LLaRP was implementing PPO training for transformer-based policies. By a "transformer-based policy", we mean a policy architecture that uses a transformer to attend to previous observations. Transformers in online RL algorithms, such as PPO, are rare. RNN-based policies are more typically used for problems where the history is important Ni et al. (2021). In this section, we include empirical analysis into important components for stable transformer-based policy training.

Prior work has demonstrated transformers working for offline RL (Xu et al., 2022; Chen et al., 2021a), but such works rely on supervised losses from static datasets. Other works explore using the transformer-based policy to collect data (Zheng et al., 2022) but use a similar offline-RL training process. Parisotto et al. (2020) use transformers for online-RL, but shows that additional transformer architecture changes are necessary for stabilizing RL training. We note that these changes are incompatible with LLM architectures, precluding the use of LLMs in their framework. We are

| | Total | Aggregated | | Train | New Scenes | Instruction Rephrasing | Novel Objects | Per Dataset Breakdown | | | | | | |
|---|---|---|---|---|---|---|---|---|---|---|---|---|---|---|
| | | Behavior Generalization | Paraphrastic Robustness | | | | | Multiple Rearrange | Referring Expressions | Context | Irrelevant Text | Multiple Objects | Spatial | Conditional Instructs |
| LLaRP | **0.32** | **0.35** | **0.30** | 0.82 | **0.79** | 0.73 | **0.82** | **0.32** | **0.21** | **0.28** | **0.23** | 0.01 | **0.05** | **0.27** |
| LSTM-Flan | 0.20 | 0.21 | 0.19 | 0.82 | 0.79 | **0.76** | 0.69 | 0.11 | 0.06 | 0.09 | 0.04 | 0.00 | 0.00 | 0.06 |
| LLaRP-Scratch | 0.17 | 0.19 | 0.16 | **0.84** | 0.79 | 0.58 | 0.60 | 0.15 | 0.01 | 0.01 | 0.17 | 0.00 | 0.02 | 0.01 |
| ZS-ChatGPT | 0.19 | 0.21 | 0.18 | 0.52 | 0.45 | 0.50 | 0.53 | 0.24 | 0.20 | 0.08 | 0.09 | **0.01** | 0.00 | 0.04 |
| ZS-LLaMA | 0.12 | 0.12 | 0.12 | 0.50 | 0.36 | 0.28 | 0.44 | 0.05 | 0.03 | 0.22 | 0.05 | 0.00 | 0.00 | 0.00 |
| ZS-Flamingo | 0.05 | 0.07 | 0.03 | 0.22 | 0.13 | 0.16 | 0.21 | 0.07 | 0.00 | 0.00 | 0.00 | 0.00 | 0.00 | 0.00 |
| LSTM-LLaMA | 0.01 | 0.00 | 0.01 | 0.29 | 0.12 | 0.07 | 0.00 | 0.00 | 0.00 | 0.00 | 0.00 | 0.00 | 0.00 | 0.00 |

Table 6: Numerical results for Language Rearrangement efficiency from Figure 11. Numbers are for 1 random seed.

able to train transformer-based policies with PPO, despite not using any of the modifications from Parisotto et al. (2020). Other works combine various transformer architectures with long-horizon RL tasks (Ni et al., 2023; Morad et al., 2023; Esslinger et al., 2022; Sopov & Makarov, 2021; Pleines et al., 2022). Unlike these works, we use billion parameter transformer models in tasks with high-dimensional visual observations.

**Effect of Batch Size on Stability**: In Figure 10a, we illustrate the importance of a large batch size for transformer-based policy training. We compare learning curves of a LSTM and transformer-based policy on 100 training episodes from the overall training dataset. Both policies are fixed to be 40M parameters and neither policy has any pre-trained LLM. The policy only takes as input the RGB visual observations and a learned embedding of the current instruction. We only analyze training performance, thus this learned embedding is sufficient for distinguishing the instructions during training. As seen from Figure 10a, the transformer runs are more unstable at lower batch sizes, with more jagged learning curves indicating drops in performance. The RNN-based policy at the same batch size does not suffer from this issue and has smooth learning for every batch size setting. Note that smaller batch sizes converge faster because they update the policy more for a fixed number of environment interactions. The transformer-based policy is also less sample efficient than the RNN-based policy for every batch size setting. We note this efficiency finding is reversed when comparing LLaRP to RNN-based approaches in Language Rearrangement.

**Effect of Pretrained LLM Weights**: We found that using pre-trained and frozen LLM weights are important for stable and fast convergence. In Figure 9a, training LLaRP-Scratch required 500M samples to converge and exhibited unstable training demonstrated by the dips in training performance. LLaRP, using the same architecture, but frozen LLM weights, learned in under 100M steps and did not have the same dips in performance during training.

## D.2 ADDITIONAL LANGUAGE REARRANGEMENT EXPERIMENTS AND ANALYSES

**Full Generalization Numbers**: In Table 5 we show the numerical results for all numbers from the paper. These include the main zero-shot generalization results from Figure 3, the scaling results from Figure 5 and the task setting results from Figure 12a.

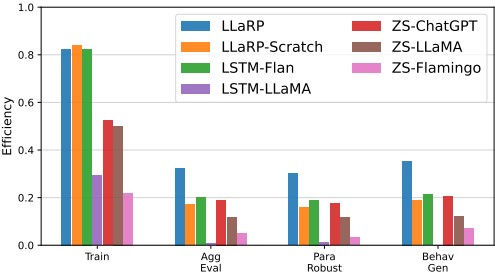

Figure 11: Language Rearrangement efficiency. Numbers are for 1 random seed. Higher numbers indicate more efficient solutions. Numbers are rescaled in $[0, 1]$ where 0 is the least efficient and 1 is the most efficient.

**Efficiency Results**: In Figure 11, we compare the efficiency of all methods in the Language Rearrangement zero-shot evaluation. We measure efficiency by the number of steps agents take to solve the tasks. Specifically we compute the efficiency of episode $i$ as $1 - \frac{n_i}{\text{max\_steps}}$ where $n_i$ is the number

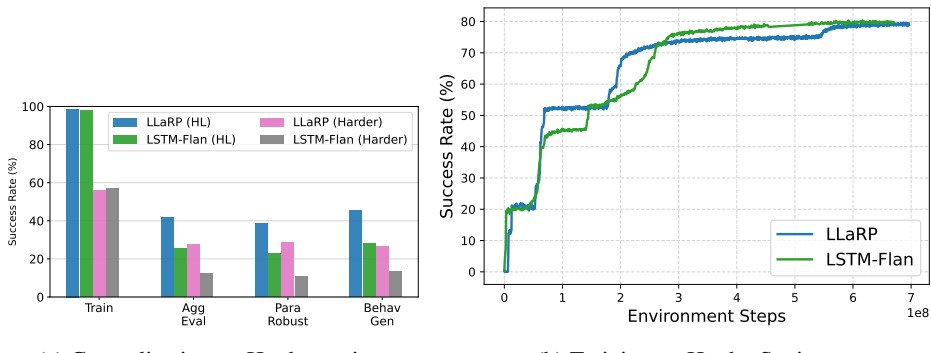

(a) Generalization on Harder setting.  (b) Training on Harder Setting.

Figure 12: Left: The zero-shot evaluation performance on Language Rearrangement where invalid actions result in immediate failure and the stop action is required. Right: Training curves under this harder setting. We enforce these termination conditions during training as well, resulting in slower training. Results in the harder task are just for 1 seed since training is slower.

of steps needed to complete episode $i$. $n_i$ is assigned max_steps if the episode was unsuccessful. We report the average efficiency over all episodes in the evaluation datasets. LLaRP is more efficient than baselines, and is over $1.5$x as efficient as the best performing baseline.

**Importance of Unfreezing LLM Weights in LSTM-Flan**: In Figure 10b we analyze the effect of freezing or unfreezing the Flan-T5 encoder weights during training. Unlike for LLaRP, keeping the LLM weights frozen results in the policy only learning the easy instructions even after 200M steps of training. We found fine-tuning the language encoder to be effective at learning the harder instructions and converging much faster to the maximum training performance.

**LLaRP Full Finetuning (LLaRP-FT)**: In Table 5, we compare the effect of not freezing the LLM in LLaRP and fine tuning it during training (LLaRP-FT). Like with training LLaRP-13b, we use model parallelism and scale training to 4 nodes (8 GPUs each). However, as seen in Figure 9b, training performance is poor. We were only able to train for 15M steps due to the slow training speeds. A larger batch size and longer training could result in LLaRP-FT working better.

### D.3   HARDER TASK VARIANT

We consider a harder variant of Language Rearrangement where invalid actions immediately end the episode and the agent must call a termination action at the end of the episode. Calling the termination action before the task is successfully completed results in a failed episode. We train LLaRP and the best performing baseline, LSTM-Flan, in this setting. We train on the same training datasets and evaluate on the same holdout datasets as in the main Language Rearrangement experiments.

We find training policies to be more difficult in this setting. As seen from the learning curves in Figure 12b, even after 700M steps of training, policies are not yet converged and still cannot solve the hardest instructions. In Figure 12a, we analyze the zero-shot performance on the Language Rearrangement evaluation datasets. Both LLaRP and LSTM-Flan suffer a drop in performance on this harder task. LLaRP still greatly outperforms LSTM-Flan in this harder setting.

### E   QUALITATIVE RESULTS

In Figure 13, we visualize success examples for LLaRP. See the figure caption for a breakdown of each of the success examples.

Below we also describe common failure modes of LLaRP on the datasets where LLaRP has less than 90% success rate.

- Context: For the instructions involving "playing a sport" and "clean a spill", the agent never picks up the desired object and instead aimlessly navigates between receptacles.

- Referring Expressions: Like in context, the agent randomly moves around for some of the instruction types like "yellow round fruit" (lemon) and "purple fruit" (plum).
- Irrelevant Instruction Text: The agent will sometimes move a wrong object never described in the instruction or distractor text.
- Multiple Rearrangements: The agent moves 2 of the 3 objects.
- Multiple Objects: The agent often fails to pick the 2nd object when it exists in the scene.
- Conditional Instructions: The agent fails to check if the fridge is open before doing the rearrangement.

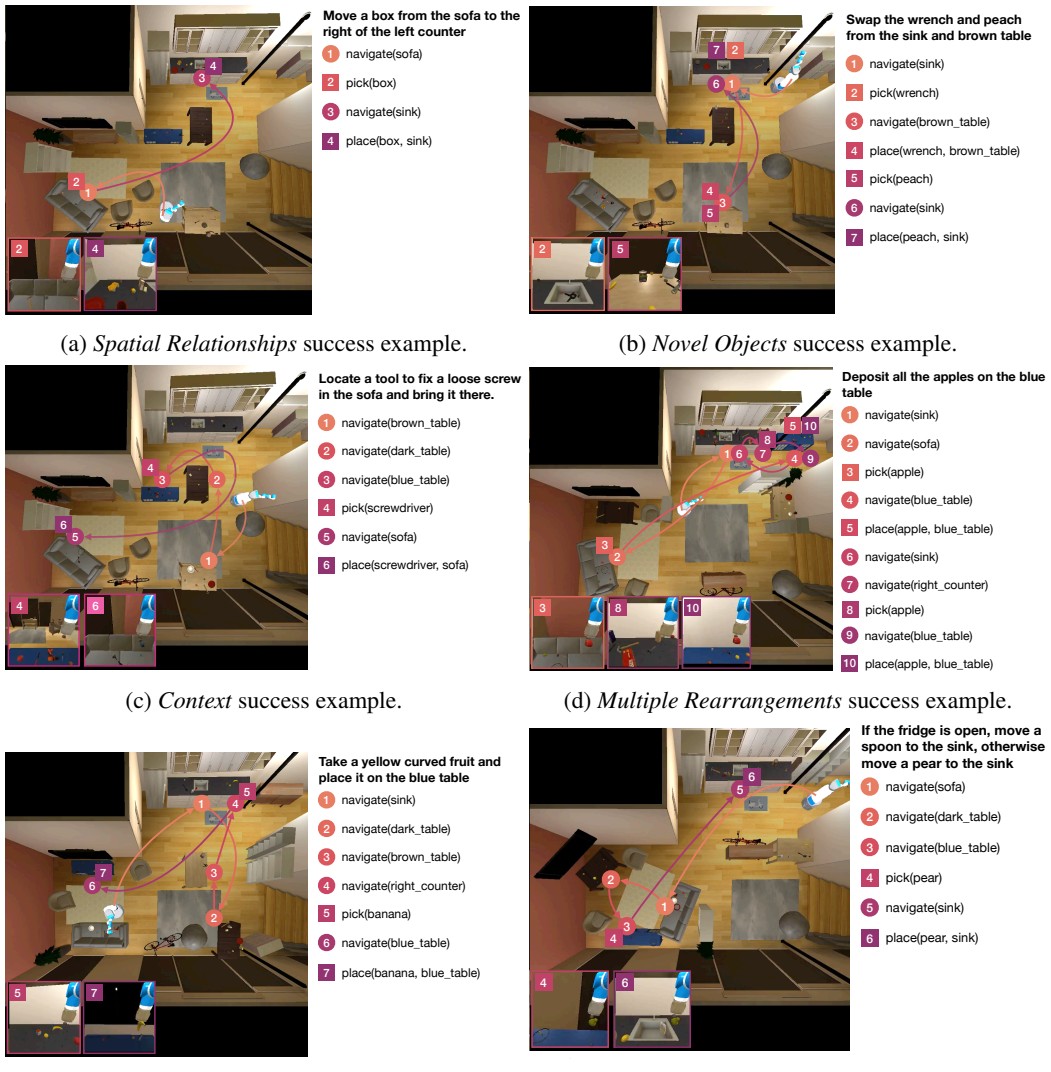

(a) *Spatial Relationships* success example.

(b) *Novel Objects* success example.

(c) *Context* success example.

(d) *Multiple Rearrangements* success example.

(e) *Referring Expressions* success example.

(f) *Conditional Instructions* success example.

Figure 13: Successful LLaRP trajectories in some of the Language Rearrangement evaluation datasets. Fig. 13a: the agent infers the sink is the receptacle to the right of the left counter. Fig. 13b: the agent has to interact with the wrench not seen before during training. It correctly picks the wrench despite not seeing this object in the context of this instruction during training. Fig. 13c: the agent infers a screwdriver is needed to satisfy the context of "a loose screw". It explores until it finds the screwdriver and then brings it to the couch. Fig. 13d: The agent explores until it finds an apple at which point it places it on the receptacle. The agent continues to explore to find if there are other apples, finds one on the right counter, and then rearranges it to the blue table. Fig. 13e: The agent explores until it sees a banana which it infers is a "yellow curved fruit". The agent then moves the banana to the blue table. Fig. 13f: The fridge is closed, so the agent is correct to move the pear to the sink. However, the agent didn't explicitly check to see if the fridge is open. This likely prevents LLaRP from achieving higher success on the *Conditional Instructions* dataset.

| | |
|---|---|
| Train | Move a 'target_object_name' from the 'source_receptacle_name' to the 'target_receptacle_name' |
| | Move a 'target_object_name' to the 'target_receptacle_name' |
| | Move the 'target_object_name' off the 'target_receptacle_name'. |
| | Place a 'object1' and a 'object2' on the 'targ_recep'. |
| | Can you swap the 'object1' and the 'object2' in the 'receptacle1' and 'receptacle2'? |
| | I accidently left the fridge open, can you close it? |
| | Can you open the fridge for me? |
| | Go to 'target_receptacle_name'. |
| | Find a 'target_object_name'. |
| Instruction Rephrasing | On the 'source_receptacle_name' there is a 'target_object_name', move it to the 'target_receptacle_name' |
| | On the 'target_receptacle_name' I need you to put a 'target_object_name' |
| | Set out a 'plate' for one person on the 'target_receptacle_name'. |
| | The 'target_receptacle_name' should be devoid of any 'target_object_name'. |
| | On the 'targ_recep', I need a 'object1' and a 'object2'. |
| | I misplaced the 'object1' on the 'receptacle1' and the 'object2' on the 'receptacle2'. Can you swap their positions? |
| | When putting away groceries, I forgot to shut the fridge. Can you help? |
| Referring Expressions | Bring a green fruit to the 'target_receptacle_name' |
| | Bring a yellow round fruit to the 'target_receptacle_name'. |
| | Bring a yellow curved fruit to the 'target_receptacle_name' |
| | Bring a round red fruit to the 'target_receptacle_name' |
| Multiple Objects | Put all the 'object_name' on the 'targ_recep'. |
| | Put all the 'object_name' from the 'receptacle1' on the 'receptacle2'. |
| Conditional Instructions | If the fridge is open move a 'target_object_name1' to the 'target_receptacle_name', otherwise move a 'target_object_name2' to the 'target_receptacle_name'. |
| | If the fridge is open move a 'target_object_name1' to the 'target_receptacle_name', otherwise move a 'target_object_name2' to the 'target_receptacle_name'. |
| Spatial Relationships | Move a 'target_object_name' from the 'source_receptacle_name' to the left of the right counter. |
| | Move a 'target_object_name' from the 'source_receptacle_name' to the right of the left counter. |
| | Move a 'target_object_name' from the 'source_receptacle_name' to the right of the TV stand. |
| Context | I want to play a sport, bring something to play with to the 'target_receptacle_name'. |
| | A screw is loose in the 'target_receptacle_name', bring something to fix it. |
| | I need to cut a piece of paper at the 'target_receptacle_name', can you bring something to help? |
| | I spilt something and need to clean it. Can you bring something to the 'target_receptacle_name' to help? |
| | Bring me something to pour hot coffee into at the 'target_receptacle_name' |
| Multiple Rearrangements | Move the 'obj1' to the 'targ_recep1', the 'obj2' to the 'targ_recep2', and the 'obj3' to the 'targ_recep3'. |
| Irrelevant Instruction Text | There's an apple on the sofa, but on the 'target_receptacle_name' I need you to put a 'target_object_name' |

Table 7: Sampling of the instruction templates for each of the task datasets. Note that for each template we include multiple phrasings. Names in backtics indicate template placeholders. Our pipeline described in Appendix B.2 automatically grounds these placeholders with feasible entity names.

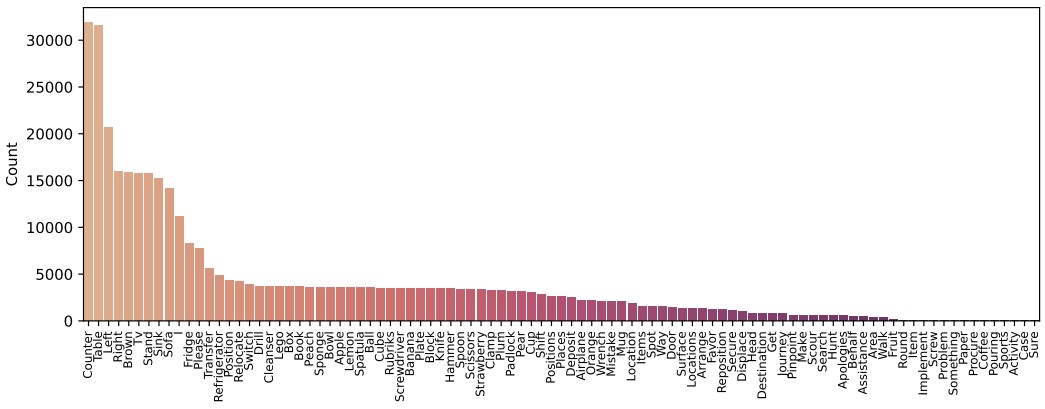

(a) Noun frequency for all instructions.

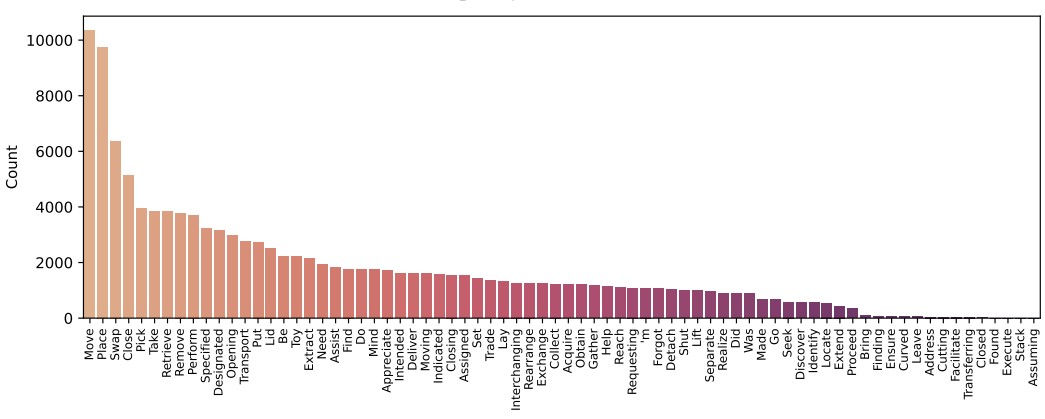

(b) Verb frequency for all instructions.

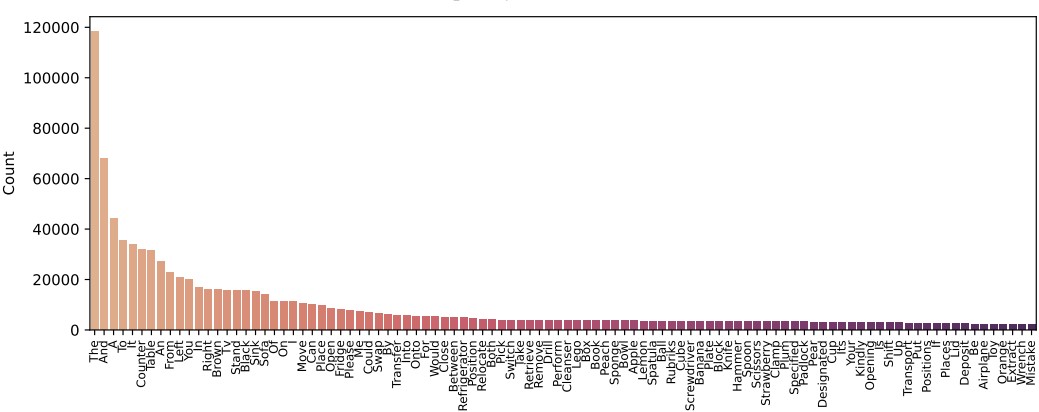

(c) Word frequency for all instructions

Figure 14: Distribution of word counts for all the instructions.

