# OpenReview forum: "Large Language Models as Generalizable Policies for Embodied Tasks"
_ICLR.cc/2024/Conference — ICLR 2024 poster_

### Official Review · Reviewer_92Kq · 2023-10-16

**Soundness:** 3 good
**Presentation:** 3 good
**Contribution:** 2 fair
**Rating:** 6
**Confidence:** 4

**Summary:**

This paper proposed a new method to adapt LLMs to embodied visual tasks, to leverage the world knowledge of LLM and achieve better generalization on new tasks.
In this method, a pre-trained frozen LLM is used to take text instructions as input. A frozen vision encoder is used to encode visual observations. A trainable MLP is used to map the output of visual encoder to the LLM input space. Finally, a few MLPs are used to map LLM output to a policy's action spaces.

**Strengths:**

The authors proposed a nice way to leverage LLM in the embodied ai tasks with visual inputs. LLM in the design are essentially producing the state representation for the action decoder, given a visual input adaptor. The total number of trainable parameters are constraints so that RL can be efficiently applied.
Empirically, the authors showed that the learned policy generalizes well to large diverse tasks. The unseen tasks are divided into two categories: paraphrastic robustness and behavior generalization to evaluate the generalization of the policy. The empirical evaluation method itself has its own value.

**Weaknesses:**

The proposed method uses MLP to map LLM output to discrete actions. Therefore, the actions and LLM outputs are not in the same space. Due to this, I am not sure if the policy can fully leverage the world knowledge from LLM. It would be better if the actions are directly produced by LLM. Of course, in that case, we may need to unfreeze the LLM weights.

**Questions:**

Why not use a VLM backbone but instead use a language-only backbone?

---

> ### Author Response · Authors · 2023-11-16
>
> We thank the reviewer for the comments and suggestions. We address the reviewer’s points below.
>
> **1. The proposed method uses an MLP to map LLM output to discrete actions. I am not sure if the policy can fully leverage the world knowledge from the LLM.**
>
> Modifying LLaRP to operate directly in the language token space rather than through the action decoder MLP is an interesting extension. We note extending LLaRP to implement this is non-trivial. Firstly, a single action in Language Rearrangement consists of at most 4 tokens (for example, “nav left counter”). This LLaRP variant requires predicting the correct sequence of 4 tokens from a vast number of tokens which decode to an invalid action. Secondly, it slows the RL process, because now the policy needs to auto-regressively predict 4 tokens per action prediction step rather than the single prediction needed in LLaRP. Despite these challenges, we agree this is an exciting direction for a follow up work and we updated the conclusion to reflect this (changes in red).
>
> We also note that we empirically demonstrate the policy is able to use the world knowledge from the LLM since LLaRP generalizes better than LLaRP-Scratch which does not have the pre-trained LLM weights.
>
> **2. Why not use a VLM backbone but instead use a language-only backbone?**
>
> It is possible to change the network weight initialization in LLaRP with VLM weights instead of using VC-1 and LLaMA as in the paper. However, we started from LLaMA and VC-1 since we found the zero-shot performance of LLaMA was higher than the Flamingo VLM (12% vs. 6% success rate).

---

> ### Author Response · Authors · 2023-11-21
>
> Dear Reviewer 92Kq, we would be grateful if you can comment on whether our response addressed your concerns or if issues remain.

---

> > ### Author Response · Authors · 2023-11-22
> > **Kind Reminder**
> >
> > Dear Reviewer,
> >
> > Thank you for appreciating the strengths of the work, and your suggestions for future extensions. Even with a direct action head, we empirically demonstrate that LLaRP leverages the knowledge encoded in the LLM across various tasks and against several classical baselines, which we believe is your only concern. Given this and also the deadline of today for a discussion, we are wondering whether you are willing to revise your rating?
> >
> > Best, Authors.

---

### Official Review · Reviewer_DZLf · 2023-10-30

**Soundness:** 3 good
**Presentation:** 3 good
**Contribution:** 3 good
**Rating:** 6
**Confidence:** 4

**Summary:**

This paper presents a method for embodied AI for object rearrangement that leverages Vision-Language Models for learning embodied policies. The paper also presents an evaluation benchmark based on a simulated environment (likely Habitat) that significantly extends the number of scenarios and natural language instructions, additionally including novel evaluation axes such as robustness to paraphrasing and robustness to alternative behaviors. The paper reports significant generalization improvements compared to baseline methods and reveals the weaknesses of alternative approaches when dealing with linguistic ambiguities and behavioral differences.

**Update after rebuttal**

After the rebuttal, and with taking the additional clarity provided by the authors during the rebuttal into account, I raise my score, since they have clarified several points on the details of their work, and its limitation. Nevertheless, I still believe that the scope of their work can be improved if they provide additional evidence for LLaRP's performance outside their own benchmark, where previous competitive work exists.

**Strengths:**

- despite existing similar benchmarks such as Habitat rearrangement task and Alfred, the proposed large-scale evaluation benchmark and their perspectives on generalization is useful for comparing embodied agents that can leverage both perception and language instructions, while working in an interactive simulation.
- extending habitat rearrangement with large number of natural language instructions enables research in this directions to be able to move beyond zero-shot or fewshot in-context agents.
- the paper compares the proposed approach that leverages finetuning the policy and image-to-language adaptive layers, to zeroshot methods and alternative sequence models, and reports significant improvements in generalisation of the aforementioned axes of evaluation.

**Weaknesses:**

- **incorrect claims and missing related work:** the paper states "To our knowledge, there are no prior work which demonstrate that the linguistic knowledge in LLMs can be used in online RL problems to improve generalization in embodied AI settings". There are many examples in the literature that actually have demonstrated that, some of which have been cited (such as ELLM, PALM-E, etc) and many were not discussed (e.g, SayCan, CodeAsPolicy, PercieverActor, HELM, ProgPrompt, EmbodiedGPT,...), though there are many examples of such methods. I recommend correcting the inaccuracies in such statements.
- **lack of details:** the main paper lacks many details such as what simulation were used, how the agent executes the skills, how the skills are defined, and what is the contribution of the defined PDDL. Although some details are provided in the supplementary material, clarifications are needed to be presented in the main paper.
- **lack of comparison to existing benchmarks:** The paper does not provide a comprehensive comparison to available benchmarks (some examples includes but not limited to Habitat rearrangement, CortexBench, AI2Thor, Behavior1k, Procthor, ALFRED). Hence, it is not clear how the new provided axes and extending instructions stand against existing work. It is quite common in the literature to dedicate a section and a table to compare various aspects of a newly proposed benchmark to existing ones from various aspects such as scale, generalization aspects, number of samples, kinds of provided data (such as language instructions) etc.
- **limited comparison to existing work:** although this paper compares to relevant approaches that have proven to be effective such as zeroshot in-context text-only, as well as encoder+LSTM, there are many seminal works that are applicable to this environment and it would bring more value to comparisons if the paper actually leverages some pre-existing methods. Examples of such methods include CodeAsPolicy (Code Gen), PercieverActor (BC), and SemanticHELM (pretrained vision and LLM+LSTM). Atari is a well-established benchmark, which has many competitive baselines, which has not been provided for comparison. Providing additional evidence on other embodied AI benchmarks where previous established baselines exists provides better comparison datapoints to compare the proposed method to the existing work.

**Questions:**

- extend related work and discuss relation to existing work that has been detailed in the weaknesses section.
- provide details that have been denoted in the weaknesses section.
- provide comparison to existing benchmarks that has been pointed out in the weaknesses section.
- extend empirical evaluations and comparison to existing work that has been described in the weaknesses section.

---

> ### Author Response · Authors · 2023-11-16
>
> We thank the reviewer for the comments and suggestions. We address the reviewer’s points below.
>
> **1. Incorrect claims and missing related work: the paper states "To our knowledge, there are no prior work which demonstrate that the linguistic knowledge in LLMs can be used in online RL problems to improve generalization in embodied AI settings". There are many examples in the literature that actually have demonstrated that.**
>
> We corrected this claim to “To our knowledge, there is no prior work which demonstrates that LLMs can be used as vision-language policies in online RL problems to improve generalization” (changes in paper in red). This updated claim is accurate and the mentioned works do not demonstrate this. We apologize for this: this was an error that played out in the final editing of our first submitted draft (please refer to the general response). We compare to the works mentioned by the reviewer in detail below:
>
> - SayCan, CodeAsPolicy and ProgPrompt are zero-shot applications of LLMs that require describing the environment in text [1-3]. LLaRP adapts the LLM for _visual_ decision-making with _online RL_. By online RL, we mean learning from interactions with the environment.
> - PercieverActor, EmbodiedGPT and PaLM-E [4-6] learn from static expert collected datasets with supervised learning. LLaRP doesn’t need expert data and instead learns with RL.
> - ELLM uses LLMs as a _reward to guide_ non-LLM policies from _text-based state descriptions_ [7]. We use the LLM directly as the decision-making policy from _visual observations_.
> - HELM uses LLMs (CLIP) as a “memory mechanism”, which we see as orthogonal from our contribution of using LLMs directly for policy decision-making [8]. Unlike LLaRP, HELM does not modify the LLM, use it for better generalization, or use it directly for decision-making.
>
> **2. Lack of task details.**
>
> We expanded Supp. A.4 to describe the simulation in detail. We also added a description of the key simulation details in a new paragraph in the main paper (Sec 4.1). Due to limited space, we described how the agent executes the skill and skill definitions in Supp. A.3 and the PDDL definition in Supp. A.1.
>
> We never claim the defined PDDL is a contribution, it is only used in the implementation of the Language Rearrangement task (which is a contribution).

---

> ### Author Response · Authors · 2023-11-16
>
> **3. Lack of comparison to existing benchmarks.**
>
> Thank you for the suggestion, we include a table highlighting the differences between Language Rearrangement and prior benchmarks below without citations and in full in Table 4, supplementary A.7.
>
> Language Rearrangement has more instructions than prior benchmarks. It has almost 18x more instructions than ALFRED (151k vs 8k) and 8x more instruction templates than CALVIN (282 vs. 35). “# instructions” refers to the number of linguistically distinct instruction types, meaning “pick the apple” and “pick the pear” are counted as two separate instructions. “# Instruction Templates” refers to distinct instructions regardless of specific entities, so the same picking example is only counted as a single instruction. Also unique about Language Rearrangement is that it has dense rewards for language specified tasks. All other tasks with language instructions only define sparse success-based rewards. Since Language Rearrangement is implemented with Habitat 2.0, it also inherits the same fast simulation speeds.
>
> | Benchmark                                         | # Instructions (Tasks)            | # Instruction Templates   | Observation Type | Reward Type     | Generalization Type                                                                              | Sim Speed steps/second | # Scenes | # Objects |
> |---------------------------------------------------|----------------------------|---------------------------|------------------|-----------------|--------------------------------------------------------------------------------------------------|------------------------|----------|-----------|
> | **Language Rearrangement (Ours)**                 | 151,000                    | 282                       | Visual           | Dense Reward    | Unseen Instructions [10 datasets from Tab.1], Unseen Scenes                  | 1400                   | 105      | 82        |
> | ALFRED                  | 8,055                      | 7                         | Visual           | Sparse Success  | Random Split                                                                                     | NA                     | 120      | 84        |
> | CALVIN                      | 400                        | 35                        | Visual           | Sparse Success  | Unseen Instructions                                                                              | NA                     | 1        | 1         |
> | ARNOLD                      | 32                         | 8                         | Visual           | Sparse Success  | Random Split                                                                                     | 200-400                | 20       | 40        |
> | CLIPort (Ravens) | 10 (+ Procedural)         | 10                        | Visual           | Sparse Success  | Unseen Objects [colors, shapes, types]                                                           | NA                     | 1        | 56        |
> | BabyAI    | Procedural                 | Procedural        | Text             | Sparse Success  | Unseen Instructions [compositions, objects, synonyms, dialects]                                  | 3000                   | NA       | 4         |
> | Habitat Rearrangement | NA                        | NA                        | Visual           | Dense Reward    | Unseen Scenes                                                                                     | 1400                   | 105      | 20        |
> | Behavior1k                  | NA                        | NA                        | Visual           | Sparse Success  | NA                                                                                               | 60                     | 50       | 5215      |
> | TDW                   | NA                        | NA                        | Visual           | Sparse Success  | Unseen Scenes                                                                                     | 15                     | 5-168    | 112       |
> | ProcTHOR                        | NA                        | NA                        | Visual           | Dense Reward    | Unseen Scenes                                                                                     | 90-180                 | 10k      | 118       |

---

> ### Author Response · Authors · 2023-11-16
>
> **4. Limited comparison to existing work.**
>
> We thank the reviewer for suggestions around additional baselines, but believe they are not suited for our problem setting.
>
> - CodeAsPolicies [2] relies on perception modules which Language Rearrangement doesn't have. This is also similar to our ZS-ChatGPT baseline which replans based on environment feedback.
> - PercieverActor [4] requires offline data for imitation learning and depth observations for constructing point clouds. Language Rearrangement assumes access to neither.
> - While Semantic HELM [8] can be combined with LLaRP and baselines to increase learning efficiency and interpretability it does not address how to generalize to unseen instructions.
>
> Our experiments compare LLaRP to three RL and three zero-shot baselines. We compare three different LLM/VLMs (LLaMA, ChatGPT, and Flamingo).
>
> **Citations**
>
> [1] Ahn, Michael, et al. "Do as i can, not as i say: Grounding language in robotic affordances." arXiv preprint arXiv:2204.01691 (2022).
>
> [2] Liang, Jacky, et al. "Code as policies: Language model programs for embodied control." 2023 IEEE International Conference on Robotics and Automation (ICRA). IEEE, 2023.
>
> [3] Singh, Ishika, et al. "Progprompt: Generating situated robot task plans using large language models." 2023 IEEE International Conference on Robotics and Automation (ICRA). IEEE, 2023.
>
> [4] Shridhar, Mohit, Lucas Manuelli, and Dieter Fox. "Perceiver-actor: A multi-task transformer for robotic manipulation." Conference on Robot Learning. PMLR, 2023.
>
> [5] Mu, Yao, et al. "Embodiedgpt: Vision-language pre-training via embodied chain of thought." arXiv preprint arXiv:2305.15021(2023).
>
> [6] Driess, Danny, et al. "Palm-e: An embodied multimodal language model." arXiv preprint arXiv:2303.03378 (2023).
>
> [7] Du, Yuqing, et al. "Guiding pretraining in reinforcement learning with large language models." arXiv preprint arXiv:2302.06692 (2023).
>
> [8] Paischer, Fabian, et al. "Semantic HELM: A Human-Readable Memory for Reinforcement Learning." Thirty-seventh Conference on Neural Information Processing Systems. 2023.

---

> ### Author Response · Authors · 2023-11-21
>
> Dear Reviewer DZLf, we would be grateful if you can comment on whether our response addressed your concerns or if issues remain.

---

> ### Comment · Reviewer_DZLf · 2023-11-21
> **Re:**
>
> Dear authors,
>
> thank you for your response.
>
> - **compare to existing benchmarks**: Table4 is indeed very insightful, but does not answer the question why LLaRP is not additionally evaluated on existing benchmarks such as BabyAI to show how they compare to existing methods that have already shown results on BabyAI. This adds more evidence for generality of the proposed approach, which currently is lacking.
>
> - **compare to existing methods**: while I appreciate the authors detailed the differences between the proposed method and the discussed existing work, I would still be interested to see how other more advanced existing methods compare to LLaRP, even though they may differ in some aspects.
>
> minor:
>
> - **claims:** the updated claim reflects the contributions more accurately. Though, some previous work could still qualify for the given definition of "LLMs used as vision-language policies in online RL problems to improve generalization".
>
> - **lack of details**: the additional details clarify some of the missing information, thank you for extending the paper. I recommend to also extent the clarifications with the differences between the current benchmark and existing ones on executing low-level policies or kinetically updating with skill post conditions, in which the latter introduces less challenges.
>
> - I recommend adding the discussions about the related work and existing benchmarks to the paper.

---

> > ### Author Response · Authors · 2023-11-22
> >
> > Dear Reviewer DZLf,
> >
> > Thank you very much for responding and engaging in a discussion! Your comments are helpful in improving our submission. We would like to address the outstanding reservations you may have.
> >
> > **Compare on BabyAI.**
> >
> > We appreciate the reviewer’s point on comparing on BabyAI, an environment that tests agents on grid-world tasks. We provide results in Language rearrangement which tests agents in 3D, visually realistic homes from egocentric RGB images. We believe Language Rearrangement has more complex environments than BabyAI in terms of visual complexity, task diversity and task realism.
> >
> > May we ask the reviewer to share their concerns of what aspects of BabyAI tests that aren’t tested in Language Rearrangement?
> >
> > **Comparison to Existing Methods.**
> >
> > Although we strive to compare against methods that can be applied in our setup, the methods that you listed cannot be compared against because they make assumptions that are not available in our setting.
> > - CodeAsPolicies does not directly accept images as input, and instead relies on predefined perception modules that we don’t have in Language Rearrangement. Hence, it isn't technically possible to be applied on Language Rearrangement, and provide a comparison.
> > - PerceiverActor requires expert imitation learning data and depth sensor input. Language Rearrangement does not provide either. As such, it is not possible to provide a comparison.
> >
> > We have done our best to compare to a variety of advanced existing methods, from zero-shot methods using the latest LLMs and VLMs, to RL baselines trained with vast amounts of experience. We would like to point out ICLR policy (Last point in https://iclr.cc/Conferences/2024/ReviewerGuide) on no requirement of comparing with contemporaneous work, which is work deemed to be too recent to compare against; defined as published after May 28, 2023. Several of the mentioned works fall into this category (CodeAsPolicies [2] presented May 29 - Jun 1, 2023, ICRA; Progprompt [3], May 29 - Jun 1, 2023, ICRA; SemanticHelm [8], Dec 10 - 16, 2023, Neurips). We understand it is a fast moving field but would like to ask the reviewer to be mindful of policies around recently published or yet to be published work.
> >
> > **Response to minor comment on claims:** We are happy that we have clarified our work’s claims, per the reviewer's suggestion, and compared to all prior work the reviewer raised. We are happy to compare to other works if the reviewer can point them out.
> >
> > **Response to minor comment on details:** We added the information the reviewer proposed on low-level skill implementation between benchmarks in Supplementary A.7. We also included the discussions about related work and existing benchmarks in Sec. 2 and Supp. A.7.
> >
> > Given that we have addressed the outstanding reservations by the reviewer, we are wondering whether the reviewer will be willing to revise their rating?

---

> > ### Comment · Reviewer_DZLf · 2023-11-22
> > **Thank you for your response**
> >
> > After the rebuttal, and with taking the additional clarity provided by the authors during the rebuttal into account, I **raise my score**, since they have clarified several points on the details of their work, and its limitation. Nevertheless, I still believe that the scope of their work can be improved if they provide additional evidence for LLaRP's performance outside their own benchmark, where previous competitive work exists.

---

### Official Review · Reviewer_Daer · 2023-11-01

**Soundness:** 3 good
**Presentation:** 2 fair
**Contribution:** 3 good
**Rating:** 6
**Confidence:** 3

**Summary:**

The paper introduces an approach that adapts pre-trained LLMs to embodied visual tasks, by leveraging the world knowledge encoded in LLMs to enhance training efficiency. The paper also introduces a new benchmark, Language Rearrangement, with 150,000 training and 1,000 testing tasks for studying language-conditioned, which contains a diverse set of language-conditioned rearrangement tasks, such as complex manipulation, navigation, and exploration tasks.

**Strengths:**

- In the context of contemporary works, such as emdodiedGTP, Eureka, RoboCat, Open X-Embodiment, etc., this work demonstrates that the LLM-based LLaRP model exhibits strong generalization capabilities. It can handle complex paraphrasing of task instructions and generalize to new tasks.
- LLaRP shows faster convergence during training compared to other baselines, indicating its sample efficiency. Scaling up the size of the underlying LLM (from 7B to 13B parameters) leads to better results, suggesting that larger LLMs enhance embodied reasoning capabilities.
- The paper provides comparisons with zero-shot baselines, showing LLaRP outperforms models that rely solely on language understanding without training. It also shows that LLaRP trained with reinforcement learning (RL) outperforms LLaRP trained with imitation learning (IL), highlighting the effectiveness of RL in this context.

**Weaknesses:**

- What distinguishes this work from [1,2, 3, 4], which also appear to emphasize the evaluation of LLMs' generalization abilities in embodied settings?
- In the current way results are presented, it is very difficult to understand the differences in model capacity across baselines and LLaRP. Please consider including a direct comparison of model capacity in Fig. 3, as well as exact numbers in the bar plots (it seems that there is sufficient space to do so given the white space surrounding the bar plots). Same for Figures 5, 7, and so on.
- The paper presents successful results but does not thoroughly explore or discuss failure cases. It would be great to have more qualitative examples that enable readers to understand the limitations of employing pretrained LLMs on embodied tasks.
- It would be valuable to gain insights into the performance of simpler baselines, such as embedding-based models like embCLIP, or baselines from the embodied rearrangement literature such as [5], in comparison to the proposed approach. Currently, all existing baselines rely on pretrained Language Models (LLMs), with varying prompts and inputs. While this dependence on LLMs is shown to yield strong results, it would be good to understand the necessity and efficiency of LLMs for the tasks at hand. In other words, it is difficult to ground the work in existing embodied literature, since it introduces a new task (language rearrangement) and a new model (LLM-based LLaRP model trained with online RL).

[1] Li, Shuang, Xavier Puig, Chris Paxton, Yilun Du, Clinton Wang, Linxi Fan, Tao Chen et. al. "Pre-trained language models for interactive decision-making." Advances in Neural Information Processing Systems 35 (2022): 31199-31212.

[2] Carta, Thomas, Clément Romac, Thomas Wolf, Sylvain Lamprier, Olivier Sigaud, and Pierre-Yves Oudeyer. "Grounding large language models in interactive environments with online reinforcement learning." ICML (2023).

[3] Xiang, Jiannan, Tianhua Tao, Yi Gu, Tianmin Shu, Zirui Wang, Zichao Yang, and Zhiting Hu. "Language Models Meet World Models: Embodied Experiences Enhance Language Models." NeurIPS (2023).

[4] Huang, Wenlong, Pieter Abbeel, Deepak Pathak, and Igor Mordatch. "Language models as zero-shot planners: Extracting actionable knowledge for embodied agents." In International Conference on Machine Learning, pp. 9118-9147. PMLR, 2022.

[5] Wijmans, Erik, Irfan Essa, and Dhruv Batra. "VER: Scaling On-Policy RL Leads to the Emergence of Navigation in Embodied Rearrangement." Advances in Neural Information Processing Systems 35 (2022): 7727-7740.

**Questions:**

- How are the current baselines chosen, e.g., what would be the reason for not comparing with an LSTM-Flamingo baseline or using other VLMs such as InstructBLIP, LLaVA, miniGPT?

---

> ### Author Response · Authors · 2023-11-16
>
> We thank the reviewer for the comments and suggestions. We address the reviewer’s points below.
>
> **1. What distinguishes this work from [1,2,3,4], which also appear to emphasize the evaluation of LLMs' generalization abilities in embodied settings?**
>
> What distinguishes our work from [1,2,3,4] is that our main contribution is to demonstrate that an LLM can be adapted with _online RL_ to serve as a generalizable _vision language_ (VLM) policy for embodied AI tasks. We apologize for the confusion here. We updated the writing in sections 1 and 2 to better reflect this (changes in red) and refer to our general response for more details.
>
> We appreciate the opportunity to talk about this main contribution w.r.t. the prior works mentioned. [1,3] primarily focus on finetuning a _text only_ LLM with supervised learning in _text-based_ observation and action spaces. LLaRP adapts an LLM with _online RL_ for _visual, instruction_ tasks directly in the environment action space. [2] also uses RL, but does so in text-based environments. We adapt LLMs to visual tasks and environment action spaces. [4] along with related works [6,7] are "zero-shot policies for interactive decision-making tasks, without task specific training, in settings where the states and action spaces are both text-based" (Sec. 2).
>
> **2. More qualitative examples for understanding limitations of employing pretrained LLMs on embodied tasks.**
>
> Thank you for the suggestion. We included failure analysis in supplementary section D to support the success qualitative examples in figure 6 and section D.
>
> **3. Valuable to gain insights into the performance of simpler baselines.**
>
> The LLaRP-Scratch baseline is a simple baseline that does not rely on LLMs (Sec. 5.1). LLaRP-Scratch shows the necessity of LLMs due to its lesser generalization performance (17% vs 42% for LLaRP in Table 2) and efficiency (it takes 500M vs. 50M steps to converge, Fig. 4a).
>
> In this rebuttal, we added a comparison to using EmbCLIP [8] in LLaRP. As with [8] we used the pre-trained CLIP-ViT-L as the visual encoder for the policy. This performed slightly worse than LLaRP, achieving 40% total success rate, vs. 42% success rate for LLaRP. VC-1 [9] also reported better performance than EmbCLIP for other Habitat tasks. We also note that [5] is a method solely for improving training wall clock time, while it could help scale all our experiments, by itself it will not improve generalization performance.
>
> **4. How are the current baselines chosen, why not compare with other VLMs?**
>
> We chose Flan for the LSTM-Flan baseline because Flan is a powerful encoder model that can embed instructions into fixed length vectors, which the LSTM then takes as input. Using a decoder-based model instead, such as Flamingo, is better suited for text generation, not encoding instructions. We show this with the LSTM-LLaMA model performing significantly worse than the LSTM-Flan model (Table 2).
>
> We used IDEFICS since, at the time of submission, it was the state-of-the-art VLM model.
>
> **Citations**
>
> [1] Li, Shuang, et al. "Pre-trained language models for interactive decision-making." Advances in Neural Information Processing Systems 35 (2022): 31199-31212.
>
> [2] Carta, Thomas, et al. "Grounding large language models in interactive environments with online reinforcement learning." arXiv preprint arXiv:2302.02662 (2023).
>
> [3] Xiang, Jiannan, et al. "Language Models Meet World Models: Embodied Experiences Enhance Language Models." arXiv preprint arXiv:2305.10626 (2023).
>
> [4] Huang, Wenlong, et al. "Language models as zero-shot planners: Extracting actionable knowledge for embodied agents." International Conference on Machine Learning. PMLR, 2022.
>
> [5] Wijmans, Erik, Irfan Essa, and Dhruv Batra. "VER: Scaling On-Policy RL Leads to the Emergence of Navigation in Embodied Rearrangement." Advances in Neural Information Processing Systems 35 (2022): 7727-7740.
>
> [6] Huang, Wenlong, et al. "Grounded decoding: Guiding text generation with grounded models for robot control." arXiv preprint arXiv:2303.00855 (2023).
>
> [7] Liang, Jacky, et al. "Code as policies: Language model programs for embodied control." 2023 IEEE International Conference on Robotics and Automation (ICRA). IEEE, 2023.
>
> [8] Khandelwal, Apoorv, et al. "Simple but effective: Clip embeddings for embodied ai." Proceedings of the IEEE/CVF Conference on Computer Vision and Pattern Recognition. 2022.
>
> [9] Majumdar, Arjun, et al. "Where are we in the search for an Artificial Visual Cortex for Embodied Intelligence?." arXiv preprint arXiv:2303.18240 (2023).

---

> ### Author Response · Authors · 2023-11-21
>
> Dear Reviewer Daer, we would be grateful if you can comment on whether our response addressed your concerns or if issues remain.

---

> > ### Comment · Reviewer_Daer · 2023-11-22
> > **Thank you for the rebuttal**
> >
> > Thank you to the authors for providing a comprehensive and well-structured rebuttal. Given the clarifications, I have decided to raise my score. However, I would like to note that despite the improvements, the technical contribution of combining LLMs with online RL, while simple and straightforward, seems to require collecting a lot of online experience (Figure 4). The raised score reflects the positive impact of the authors' responses on clarity and novelty but acknowledges the reservations regarding the technical contribution and computational efficiency (which could be improved by future work).

---

### Official Review · Reviewer_vC8Q · 2023-11-01

**Soundness:** 3 good
**Presentation:** 3 good
**Contribution:** 2 fair
**Rating:** 5
**Confidence:** 3

**Summary:**

This paper introduces Large Language model Reinforcement Learning Policy, which adapts a pre-trained LLM to take text instruction and egocentric observations as inputs, and output actions. Fixing the body of LLM, adapters are trained using the standard reinforcement learning algorithm. The paper is also provided a new benchmark called language rearrangement, consisting of 150,000 training sets. They empirically show the benefits of the proposed method, comparing with several simple baselines including zero-shot LLM, zero-shot VLM, and LSTM policy in conjunction with T5 and LLaMA.

**Strengths:**

(1) The paper is well written and easy to follow.

(2) The method is simple and straightforward extension of prior studies but seems to provide solid performance gain.

**Weaknesses:**

(1) Technical novelty is not high. Using VLMs to control a task is not new, e.g., PaLM-E and RT-2 as discussed in this paper. It is true these two method does not employ reinforcement learning, but architectural or technical difference is slight and is not well discussed in the paper.

(2) While the one of the main claim or implication of the paper might be linguistic knowledge in LLMs can be used in online RL, but the statement itself is already validated in prior studies, e.g. [1]. While it is true that [1] focus on the textual environment, but there are no discussion how this paper extends the prior understanding on what kind of knowledge is encoded in the LLM and what is not.

[1] Grounding Large Language Models in Interactive Environments with Online Reinforcement Learning

(3) The paper lacks in depth analysis of the failure case, which might be important to dig in the internal knowledge of LLM.

**Questions:**

See Weakness section.

---

> ### Author Response · Authors · 2023-11-16
>
> We thank the reviewer for the comments and suggestions. We address the reviewer’s points below.
>
> **1. Technical novelty is not high. Using VLMs to control a task is not new.**
>
> We disagree and believe our work is novel in training VLMs _with reinforcement learning_ (RL) to control a task. While other works have trained VLMs from LLMs in offline settings, no other works have trained VLMs with online RL (i.e., without expert demonstrations). RT-2 and PaLM-E are _offline_: they are given 130,000 human teleoperated demonstrations collected over almost 2 years as training data [2,3]. LLaRP only has access to interaction data collected from its current policy (it uses PPO, which is on-policy), and we demonstrate that, by adapting LLMs in this way, LLaRP achieves generalizable decision-making.
>
> The architectural differences from online LLaRP and prior offline VLM decision-making models are rooted in the relative differences between online vs offline learning. RT-2 and PaLM-E use expert demonstrations, so their training is similar to supervised learning: they take a single image as input along with the task specification, and fine-tune the LLM to generate the text-based action instructions from the expert demonstrations. LLaRP does not have access to this expert data, and so trains using an online RL algorithm. Exploration is a critical capability for online learning, so in order to help in this regard, LLaRP takes as input all observations from the current episode (up to 32). Since LLaRP does not have access to expert actions (e.g., as text or otherwise), we add an action decoder module and train it with RL using the reward signal from the environment.
>
> Finally, beyond technical novelty, a useful contribution of our paper is the empirical analysis of the generalization capabilities of using an LLM as a VLM policy trained with online RL, as we explain in the next response.
>
> **2. While the one of the main claim or implication of the paper might be linguistic knowledge in LLMs can be used in online RL, but the statement itself is already validated in prior studies**
>
> This was a writing error on our part since our main claim is about adapting LLMs for _vision language_ policies with online RL. We mistakenly omitted that the novelty is specific to vision-based policies with online RL in the related work section. The reviewer is correct that [1] shows LLMs for online RL in textual environments. We updated the related work writing to address this and the connection to [1] (changes in red). Thank you for bringing this to our attention.
>
> We apologize for the confusion, but the corrections are minimal and do not change our main contribution of adapting LLMs for VLM policies with online RL. [1] also uses LLMs in online RL, but focuses on environments with text-inputs and text-outputs, thus this method does not apply to the visual decision-making in our setting. The LLaRP architecture is also different from [1] since we train an observation encoder and action decoder module, which are additional components to operate from visual observations and general action spaces.
>
> The empirical analysis in Sec. 5 also contains novel insights about LLMs for embodied AI, which itself is a contribution. We benchmark a wider variety of baselines than [1]. We include zero-shot baselines with powerful LLMs (ZS-ChatGPT, ZS-Flamingo, ZS-LLaMA), while [1] only compares to Flan-T5. We train policies for over 100x more experience (200M vs 1.5M steps). We show generalization to 1,000 new instructions across 10 distinct splits. [1] only shows generalization to new objects, compositions, synonyms, and translation (French vs. English). Unlike [1] we also show the efficiency of fine tuning to new downstream tasks (Fig 4b).
>
> **3. The paper lacks in depth analysis of the failure case.**
>
> Thank you for the suggestion. We included failure analysis in supplementary section D to support the success qualitative examples in figure 6 and section D.
>
> **Citations**
>
> [1] Carta, Thomas, et al. "Grounding large language models in interactive environments with online reinforcement learning." arXiv preprint arXiv:2302.02662 (2023).
>
> [2] Brohan, Anthony, et al. "Rt-2: Vision-language-action models transfer web knowledge to robotic control." arXiv preprint arXiv:2307.15818 (2023).
>
> [3] Driess, Danny, et al. "Palm-e: An embodied multimodal language model." arXiv preprint arXiv:2303.03378 (2023).

---

> ### Author Response · Authors · 2023-11-21
>
> Dear Reviewer vC8Q, we would be grateful if you can comment on whether our response addressed your concerns or if issues remain.

---

> > ### Author Response · Authors · 2023-11-22
> > **Kind Reminder**
> >
> > Dear Reviewer vC8Q,
> >
> > Thank you for appreciating the simplicity and performance strengths of the approach.
> >
> > We believe we address your concerns of comparison to PALM-E and RT-2 that, contrary to us, are not online RL approaches (and also according to ICLR guidelines https://iclr.cc/Conferences/2024/ReviewerGuide, last point, are considered contemporaneous and should not be used to evaluate novelty); [1] not using visual inputs, contrary to us, that is hugely important for embodied AI; Language Rearrangement providing the most thorough analysis across 9 different dimensions; and a newly compiled failure analysis thanks to your suggestion.
> >
> > Given the deadline for discussion of Nov 22nd, we would like to kindly urge the reviewer to engage in a discussion and if satisfied to revise their rating.
> >
> > Thank you, Authors.

---

### Author Response · Authors · 2023-11-16
**Response to all Reviewers**

We thank the reviewers for their useful and insightful feedback. We address the shared reviewers’ concerns about claims about prior work (vC8Q,Daer,DZLf). Our key contributions are twofold:

**1. Adapting an LLM with online reinforcement learning to serve as a VLM policy which generalizes to unseen instructions.**
Due to multiple concerns (vC8Q, Daer, DZLf) of this first contribution relative to prior works, we have better clarified that this key contribution is specific to  _vision-based_ (VLM) policies with _online RL_ in the introduction of the text (correction in red). We are the first to do this: prior works do have related contributions, but either on text-only environments, supervised learning from expert collected demonstrations, or zero-shot applying LLMs without adaptation.

We acknowledge an error in our claim, “there are no prior work which demonstrate that the linguistic knowledge in LLMs can be used in online RL problems…”  in the prior version of the text (DZLf, vC8Q), the last paragraph of the related works. We have made the following correction: “there is no prior work which demonstrates that LLMs can be used as vision-language policies in online RL problems…”.

This contribution is significant as it expands our understanding of how LLMs can be applied to important embodied settings: to online settings that do not have access to expert data, to tasks that require vision and non-linguistic actions, and to tasks that have not been seen during training without further adaptation. We appreciate the reviewers’ comments and corrections, and we would be happy for any further suggestions the reviewers might have.

This contribution also contains useful technical novelties and other empirical contributions. We demonstrate a novel and practical way to adapt LLMs for embodied visual tasks using online RL: instead of fine-tuning the LLM, we show that we can add an observation encoder and action decode modules to a frozen LLM and train these added modules with online RL. We also provide in-depth empirical analysis such as scaling results, IL vs. RL comparison, and RL continual learning, a set of useful contributions that have not been provided in prior work on adapting LLMs for online RL.

**2. The Language Rearrangement benchmark.** (DZLf, 92Kq) We introduce a new benchmark for studying generalization to new language instructions in embodied AI. This benchmark has 18x more training instructions than prior work and 10 generalization splits. To clarify the contribution of Language Rearrangement over prior work, we added a new Table 4 to Supp. A.7.

We provide more details in the responses below.

---

> ### Public Comment · ~Sukai_Huang1 · 2025-03-20
> **Concerns about related work**
>
> Hi authors,
> I found that the work from Xinghang Li, Minghuan Liu, Hanbo Zhang, Cunjun Yu, Jie Xu, Hongtao Wu, Chilam Cheang, Ya Jing, Weinan Zhang, Huaping Liu, Hang Li, Tao Kong:
> Vision-Language Foundation Models as Effective Robot Imitators. ICLR 2024
>
> looks similar to yours, do you consider both works are tackling the same research problem?

---

### Meta-Review · Area_Chair_CURh · 2023-12-20

**Metareview:**

The paper presents a method for adapting large language models to embodied visual tasks - particularly in online reinforcement learning setting - and highlights effectiveness of LLMs for improved generalization to new tasks. The reviewers appreciated the clear presentation and the approach of integrating LLMs with visual inputs for embodied AI tasks. While there were concerns about the technical novelty and the depth of analysis in relation to prior studies, the strong empirical results and the introduction of a new benchmark make this paper a valuable contribution to the community. For these reasons, I recommend acceptance.

**Justification For Why Not Higher Score:**

Limited novelty in relation to existing work.

**Justification For Why Not Lower Score:**

Strong empirical performance and new benchmark contributions.

---

### Decision · Program_Chairs · 2024-01-16

Accept (poster)